# Intrinsic connectivity reveals functionally distinct cortico-hippocampal networks in the human brain

Alexander J. Barnett[1]*, Walter Reilly[1], Halle R. Dimsdale-Zucker[2], Eda Mizrak[1,3], Zachariah Reagh[1,4,5], Charan Ranganath[1]

**1** Center for Neuroscience, University of California at Davis, Davis, California, United States of America,
**2** Department of Psychology, Columbia University, New York, New York, United States of America,
**3** Department of Psychology, University of Zurich, Zürich, Switzerland, **4** Department of Neurology,
University of California at Davis, Sacramento, California, United States of America, **5** Department of
Psychological and Brain Sciences, Washington University in St. Louis, St. Louis, Missouri, United States of
America

* ajbarnett@ucdavis.edu

for Systems Neuroscience, NORWAY

**Data Availability Statement:** Data and code used
to produce figures available at https://github.com/
ajbarn/hippo_nets.

## Abstract

Episodic memory depends on interactions between the hippocampus and interconnected
neocortical regions. Here, using data-driven analyses of resting-state functional magnetic
resonance imaging (fMRI) data, we identified the networks that interact with the hippocampus—the default mode network (DMN) and a "medial temporal network" (MTN) that included
regions in the medial temporal lobe (MTL) and precuneus. We observed that the MTN plays
a critical role in connecting the visual network to the DMN and hippocampus. The DMN
could be further divided into 3 subnetworks: a "posterior medial" (PM) subnetwork comprised of posterior cingulate and lateral parietal cortices; an "anterior temporal" (AT) subnetwork comprised of regions in the temporopolar and dorsomedial prefrontal cortex; and a
"medial prefrontal" (MP) subnetwork comprised of regions primarily in the medial prefrontal
cortex (mPFC). These networks vary in their functional connectivity (FC) along the hippocampal long axis and represent different kinds of information during memory-guided decision-making. Finally, a Neurosynth meta-analysis of fMRI studies suggests new hypotheses
regarding the functions of the MTN and DMN subnetworks, providing a framework to guide
future research on the neural architecture of episodic memory.

## Introduction

Episodic memory allows us to relive past events that happened at a particular place and time
[1]. Most research investigating the neurobiology of episodic memory retrieval has focused on
the hippocampus and medial temporal lobe (MTL) cortex [2–4], but it is generally assumed
that the hippocampus supports memory by integrating information represented across distributed areas in the neocortex [5]. Consistent with this idea, functional magnetic resonance imaging (fMRI) studies of memory retrieval have shown activity and connectivity in a cortico-hippocampal network composed of medial and lateral parietal, lateral temporal, and medial

**Funding:** This work was supported by the Natural Sciences and Engineering Research Council of Canada (Postdoctoral Fellowship awarded to AB), the National Science Foundation (Graduate Research Fellowship awarded to WBR), the Office of Naval Research (awarded to CR, ONR Grant N00014-15-1-0033, and N00014-17-1-2961), and the National Institute on Aging (awarded to ZMR, T32AG050061). The funders had no role in study design, data collection and analysis, decision to publish, or preparation of the manuscript. Any opinions, findings, and conclusions expressed in this material are those of the author(s) and do not necessarily reflect the views of the Office of Naval Research or the U.S. Department of Defense.

**Competing interests:** The authors have declared that no competing interests exist.

**Abbreviations:** AT, anterior temporal; CSF, cerebrospinal fluid; DMN, default mode network; EPI, echo-planar imaging; FC, functional connectivity; FD, frame displacement; fMRI, functional magnetic resonance imaging; FWE, family-wise error; MP, medial prefrontal; mPFC, medial prefrontal cortex; MTL, medial temporal lobe; MTN, medial temporal network; PCC, posterior cingulate cortex; PM, posterior medial; ROI, region of interest; tSNR, temporal signal-to-noise ratio; vmPFC, ventromedial prefrontal cortex.

prefrontal (MP) neocortical regions along with the MTL [6–10], and lesion studies have shown that damage within this network can cause amnesia [11].

This distributed set of cortical regions that is recruited during episodic retrieval overlaps extensively with regions that comprise the default mode network (DMN). The DMN is a large-scale network that is typically identified in studies that use intrinsic functional connectivity (FC) analysis of fMRI data [12], and some studies suggest that the DMN can be differentiated into different subnetworks [13–15]. One approach to partition the DMN has been to view DMN subnetworks as an extension of the connectivity differences within the MTL [16–18]. Guided by neuroanatomical studies of MTL connectivity in rodents and nonhuman primates [19–22], studies of intrinsic FC in fMRI data [23–26] have differentiated between cortical regions that preferentially affiliate with the perirhinal cortex and anterior hippocampus and cortical regions that preferentially affiliate with the parahippocampal cortex and posterior hippocampus [27,28]. For example, Libby and colleagues [24] demonstrated that the parahippocampal cortex has relatively higher FC with a network of posterior medial (PM) cortical regions, including the posterior cingulate and retrosplenial cortex and posterior hippocampus, whereas the perirhinal cortex has relatively higher connectivity with a network of anterior temporal (AT) cortical regions, including the orbitofrontal and temporopolar cortex and anterior hippocampus. Drawing on these findings, Ranganath and Ritchey [29] reviewed evidence converging on the idea that PM and AT networks support different aspects of memory-guided behavior. This "PM/AT framework" has provided a valuable framework for interpreting memory phenomena in imaging [9,30–32], stimulation [33,34], and disease [35–37]. However, recent work has come to question the homogeneity of the PM network [38], with several studies highlighting a dissociation between parahippocampal cortex, retrosplenial cortex, and precuneal cortex from posterior cingulate cortex (PCC) [39,40] under conditions that seem to tax perceptual, relative to abstract event processing, or scene relative to face processing.

A second approach to segmenting the DMN has used data-driven methods, and these studies have also found partially conflicting network delineations [15,41,42]. These studies have generally partitioned the DMN into 3 subnetworks: an MTL subnetwork, a midline subnetwork of posterior cingulate and medial prefrontal cortex (mPFC), and a third network composed of dorsal mPFC, lateral temporal, and ventrolateral prefrontal cortex, but have focused largely on cortical regions. It is unclear, however, whether these data-driven, cortical network parcellations are relevant to understanding episodic memory or task-related recruitment of cortico-hippocampal networks [43,44].

In the present study, we sought to use the data-driven approaches previously used to characterize large-scale cortical networks to provide a comprehensive characterization of the cortico-hippocampal networks that contribute to episodic memory. Using a whole-brain, data-driven approach to examine FC in resting-state fMRI, we sought to (1) identify and partition the DMN into subnetworks and examine whether these subnetworks converge with the PM/AT framework; (2) examine the connectivity of the hippocampus to the identified networks; and (3) determine whether these cortico-hippocampal networks play different roles in episodic memory.

## Results

### Data-driven partition of neocortical networks

Our first objective was to use a data-driven approach to identify large-scale resting-state networks. Using 25 minutes of resting-state fMRI data acquired from 40 participants, we partitioned the brain into canonical resting networks using state-of-the-art techniques [45]. We extracted the average confound-corrected time series from each region in a recently published

atlas of the human neocortex [46] and computed Fisher z-transformed Pearson correlations between the time series of each cortical region and every other cortical region. We then created a group-averaged FC matrix to be used for community detection. The Louvain community detection algorithm [47] was run for 1,000 iterations, tuning the resolution (how large or small communities might be) across a range of resolution parameters to identify a partition solution that showed both high network modularity (i.e., higher within community connectivity than would be expected by chance) and high network grouping consistency (quantified using the z-Rand index [45,48]). The solution was also subject to a qualitative criterion that the partition should separate the primary sensory networks [45]. The resolution parameter of gamma = 2.005 satisfied our criteria and produced a network partition (Fig 1) that corresponds closely with previously reported partitions in other recent studies [45,49,50]. This partition identified the DMN, but, interestingly, some areas (such as the parahippocampal cortex, precuneus, and perirhinal cortex) that are often identified as part of the DMN were grouped within a separate

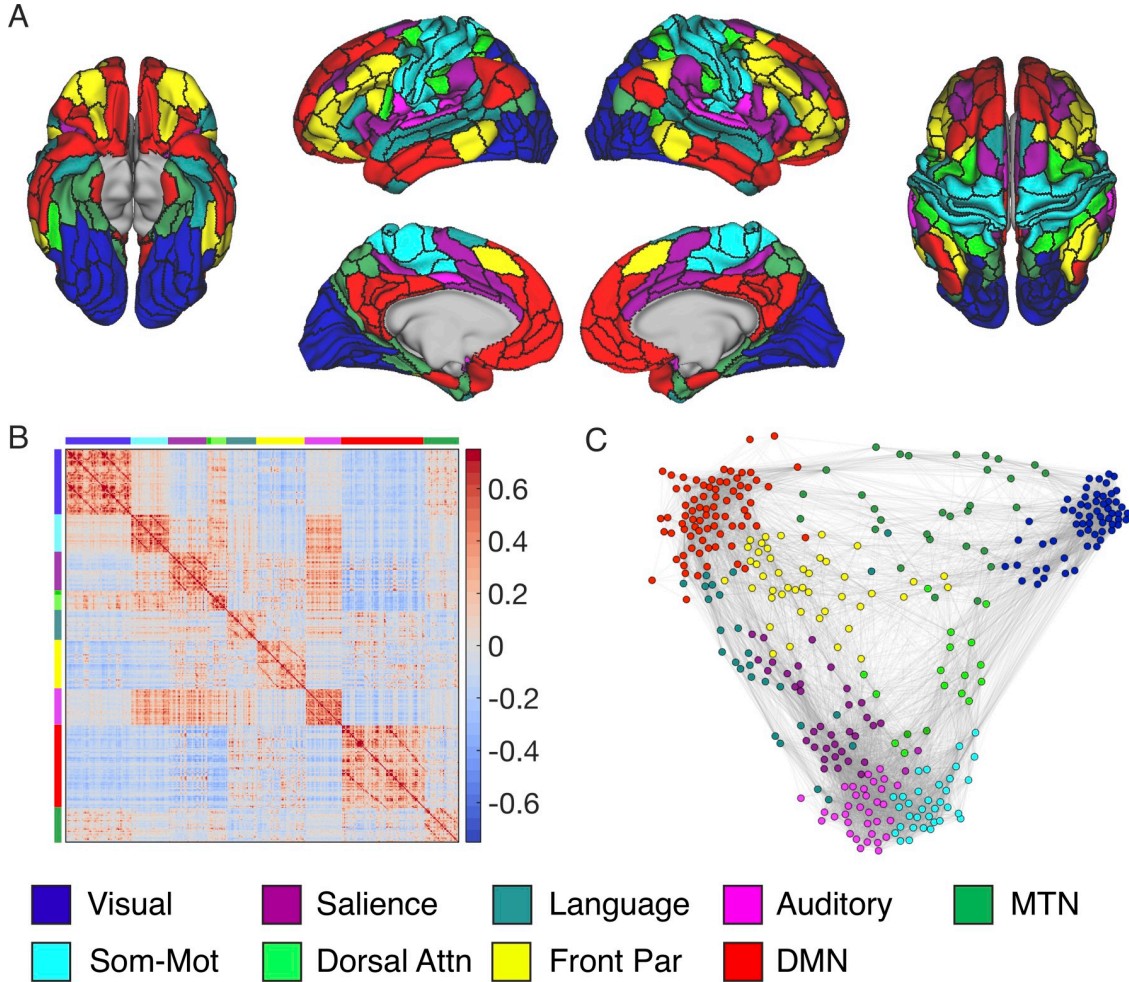

**Fig 1. Louvain community detection identifies large-scale resting-state networks. (A)** Inflated cortical surface, colored according to community membership. **(B)** Connectivity matrix reordered by community to demonstrate the community structure of the group-averaged brain. Colors along the axis demonstrate which rows/columns belong to a given community. Color bar represents Fisher Z-transformed correlation values. **(C)** Force-directed graph of the group-averaged networks color coded by community membership of the selected partition created using the ForceAtlas2 algorithm [54]. Attn, attention; DMN, default mode network; Front Par, frontoparietal; MTN, medial temporal network; Som-Mot, somatomotor. Data can be found at https://github.com/ajbarn/hippo_nets.

network that included MTL and parietal regions. This set of regions overlaps with what has been previously described as the MTL network [13], or the contextual association network [51], and coactivates particularly during recall of places [52,53]. Here, we will describe this network as the medial temporal network (MTN).

To provide internal validation of our partition, we split our dataset into 2 equal-sized subsamples and repeated the community detection procedures. We observed very high consistency across the 2 split-halves, with an adjusted z-Rand index of z = 172, $p < 0.0001$, and 95% of regions were given the same label in both halves comparable to the split-half analysis performed by Ji and colleagues [45] (split-half results shown in S1 Fig). Importantly, we observed the same expression of resting networks, including the DMN and MTN.

To provide external validation of our partition, we acquired an openly available resting-state fMRI dataset from OpenNeuro.org acquired on a 3T scanner (https://openneuro.org/datasets/ds000243/versions/00001). We used 69 participants from this dataset following preprocessing and data quality checks. The total scan length, per subject, was 12.8 minutes on average, and methodological details for data collection have been reported in previous publications [52,53]. Community detection was, again, repeated on this secondary dataset following identical preprocessing procedures, and we observed comparable delineations of these canonical networks (shown in S2A Fig). Between the primary and secondary datasets, we observed strong partition consistency with an adjusted z-Rand index of z = 150, $p < 0.0001$, and 93% of regions were given the same label in both datasets. The only major difference in network parcellations across the 2 datasets was that, in the secondary dataset, temporal polar regions differentiated into a single community that was disconnected from all of the other networks. We observed that these regions had low FC (S2B Fig) and low temporal signal-to-noise ratio (tSNR) in the secondary dataset (see S3 Fig for comparison of tSNR between datasets). Accordingly, there is good reason to believe that the isolated network of temporal polar regions in the secondary dataset is related to poor data quality in the AT lobes. Given the results of the split-half analysis reported above and the fact that our acquisition sequence showed superior data quality in the medial and AT lobes, we can be confident in the identification of communities in the AT region in the University of California, Davis dataset.

We next interrogated the FC between the networks defined above and the hippocampus. We segmented the hippocampus in each participant using FreeSurfer v6.0 (http://surfer.nmr.mgh.harvard.edu/ [55]) and then manually segmented it further into anterior and posterior regions, as these subdivisions of the hippocampus are known to have somewhat different functional properties [16,20]. The posterior hippocampus was defined as all the hippocampus posterior to the last slice of the uncal apex [16]. We calculated the connectivity of the hippocampus to every cortical region in the atlas and averaged together FC weights of cortical regions within the same network based on network affiliations. This was done for each participant. Moreover, 1-sample $t$ tests revealed that 2 networks were functionally connected to every hippocampal region of interest (ROI) in our sample—the DMN and the MTN (DMN range: $t(38) = 10.7$ to $12.9$, all $p < 0.001$; MTN range: $t(38) = 4.2$ to $9.3$, all $p < 0.01$). The language network and somatomotor network both showed significant anterior, but not posterior hippocampal connectivity (see S1 Table). The robust connectivity of the MTN and DMN to the hippocampus corroborates previous investigations of hippocampal connectivity [56].

## Network analyses support an interfacing role of the MTN

A recent review discussing the heterogeneity within the PM network suggested that parahippocampal and lateral parietal cortex interface with lower-level sensory cortex in service of higher-order feature representation (i.e., perspective invariant representation of items and spaces)

prior to binding in the hippocampus [29,38,57]. They also note that these regions interface with other DMN regions such as the mPFC and PCC (regions found in the DMN, here) that track long timescale event structure and provide conceptual knowledge pertaining to event schemas [38,58]. Thus, the MTN may serve as a bridge between external, low-level perception in the visual network and the hippocampus and DMN. By plotting the structure of the network connections in Fig 2A (left), we do indeed see that the MTN shows FC with both the DMN, hippocampus, and the visual network. Treating the brain as an abstract graph [59], we sought to test the hypothesis that the MTN mediates network communication between the visual network and the DMN and hippocampus. To do so, we used graph theory to calculate the shortest path length between the visual network and the DMN and hippocampus. Path length refers to the shortest number of links that needs to be traversed to link one node in a network to a target node [59]. If 2 regions are directly connected by a strong link, they will have a short path length, whereas if they are indirectly connected via multiple intermediate nodes, then they would have a longer path length. To examine the influence of the MTN on internetwork communication between the DMN and visual network, we compared the path length between the visual network and DMN when the MTN was removed relative to the path length when we removed all other networks. If the MTN mediates the connectivity between the visual network

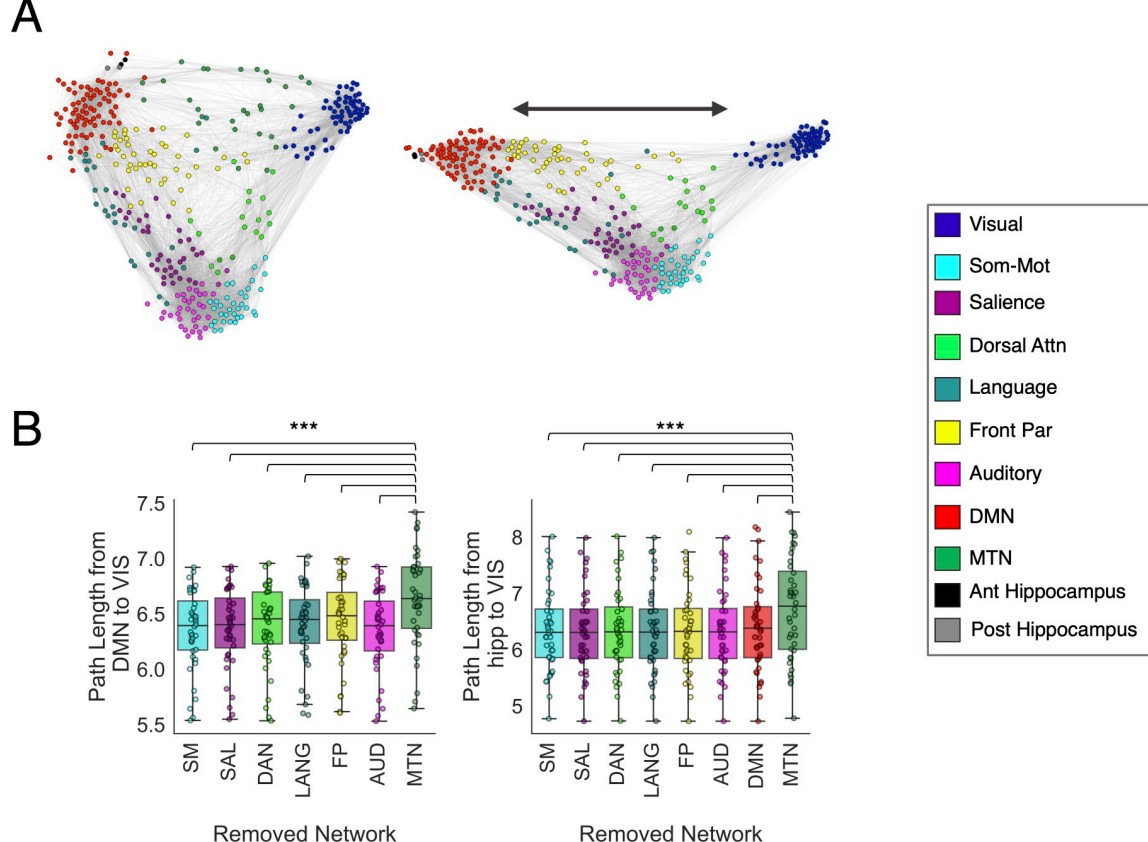

**Fig 2. Removal of the MTN disproportionately decreases the network connectivity of the DMN and visual network. (A)** Force-directed graph of the group-averaged networks color coded by community membership of the selected partition created using the ForceAtlas2 algorithm [54] with the MTN present (left) and with the MTN excluded (right). **(B)** Boxplots displaying the path length between the DMN and visual network or the path length between the hippocampi and visual network following removal of each network. *** $p < 0.001$. Ant, anterior; Attn, attention; AUD, auditory; DAN, dorsal attention network; DMN, default mode network; FP, frontoparietal; LANG, language; MTN, medial temporal network; Post, posterior; SAL, salience; SM, somatomotor; Som-Mot, somatomotor; VIS, visual. Data can be found at https://github.com/ajbarn/hippo_nets.

and DMN and hippocampus, then removal of the MTN should disproportionately increase the average path length between visual network and DMN over and above any path length increases caused by removal of all other networks. This was done at the individual subject level to determine the significance of the effect and is similar to methods used by [60].

We found that, across subjects, removal of the MTN led to a disproportionately large increase in DMN–visual network path length compared to removal of any other network (all $t(39) > 6.7$, $p < 0.001$; Fig 2B), indicating that the MTN plays a critical role in cross-network communication between the visual network and DMN. Importantly, this effect cannot be explained simply by the number of nodes removed from the whole network, as the MTN has fewer nodes (30 nodes) than the somatomotor (34 nodes), salience (35 nodes), frontoparietal (44 nodes), and auditory (33 nodes) networks. To further confirm this, we randomly removed 30 nodes from each subject's network (excluding DMN and visual nodes) and recalculated the path length values. This was performed 1,000 times, and we compared the DMN–visual network path length of the true removal of the MTN nodes compared to the distribution of DMN–visual network path length following random node removals. We observed that the MTN removal still resulted in greater average path length between the DMN and visual network relative to removal of random nodes, $t(39) = 9.4$, $p < 0.001$.

However, another explanation for this effect is that the MTN is spatially the most proximal to both the visual network and DMN within the brain. Thus, we identified a set of 32 non-MTN regions that was matched for Euclidean distance to both the visual and DMN compared to the MTN. Removal of the MTN nodes still resulted in a significantly higher path length between the DMN and visual network compared to removal of these spatially proximal matched control regions, $t(39) = 5.5$, $p < 0.001$. Further, removal of these control regions was not significantly different in terms of the resultant path length compared to the frontal parietal network removal, $t(39) = 0.7$, $p = 0.5$, although it was significantly higher than removal of the other networks (all $t(39) > 3.2$, all $p \leq 0.001$). Another possible explanation is that due to the high level of connectivity between the DMN and MTN, removal of the MTN will nonspecifically lead to a disproportionate increase in path length of the DMN to all other networks. However, the path length effects caused by removal of the MTN were specific to the DMN–visual network path lengths (S4 Fig).

As mentioned above, it is well known that MTL regions such as the perirhinal and parahippocampal cortex provide the hippocampus with higher-order representations of objects and scenes [4,61]. Therefore, we repeated the previous analysis, this time examining the effects of MTN removal on the path length between the hippocampus and visual network. We observed that removal of the MTN led to greater path length between the hippocampus and visual network compared to removal of any other networks (all $t(39) > 6$, $p < 0.001$; Fig 2B). Again, MTN removal led to disproportionately higher path length than removal of random nodes ($t(39) = 3.15$, $p = 0.003$). To account for spatial proximity, as above, we identified a set of 31 non-MTN regions that was matched for mean Euclidean distance to the hippocampus and visual network relative to the MTN. Removal of these spatial control nodes was not sufficient to account for the effects of MTN removal, as removal of the MTN still led to significantly greater increase in path length, $t(39) = 7$, $p < 0.001$. This effect was specifically between the hippocampus and visual network path length, rather than leading to disproportionately higher path length between the hippocampus and any other network (S5 Fig).

## Partitioning the DMN

We next sought to identify whether the DMN could be broken up into subnetworks using similar methods that we used in our whole-brain network partition. We selected the DMN regions

identified in our whole-brain partition and performed Louvain community detection on the functional connections between the regions in the network. This was done for 1,000 iterations across a range of resolution parameters, and we again selected the partition yielding the highest modularity and stability. The solution with the highest modularity-weighted stability produced a partition of the DMN with 3 communities which we refer to as the PM subnetwork, the AT subnetwork, and the MP subnetwork. The PM subnetwork and AT subnetwork were named as such based on their resemblance to previous spatial maps of these networks [24]. The PM subnetwork included PCC, anterior retrosplenial cortex, posterior angular gyrus, and dorsal prefrontal cortex. The AT subnetwork included temporopolar cortex, lateral orbitofrontal cortex, temporoparietal junction, and dorsomedial prefrontal cortex. The MP subnetwork included mPFC regions and the entorhinal cortex (Fig 3).

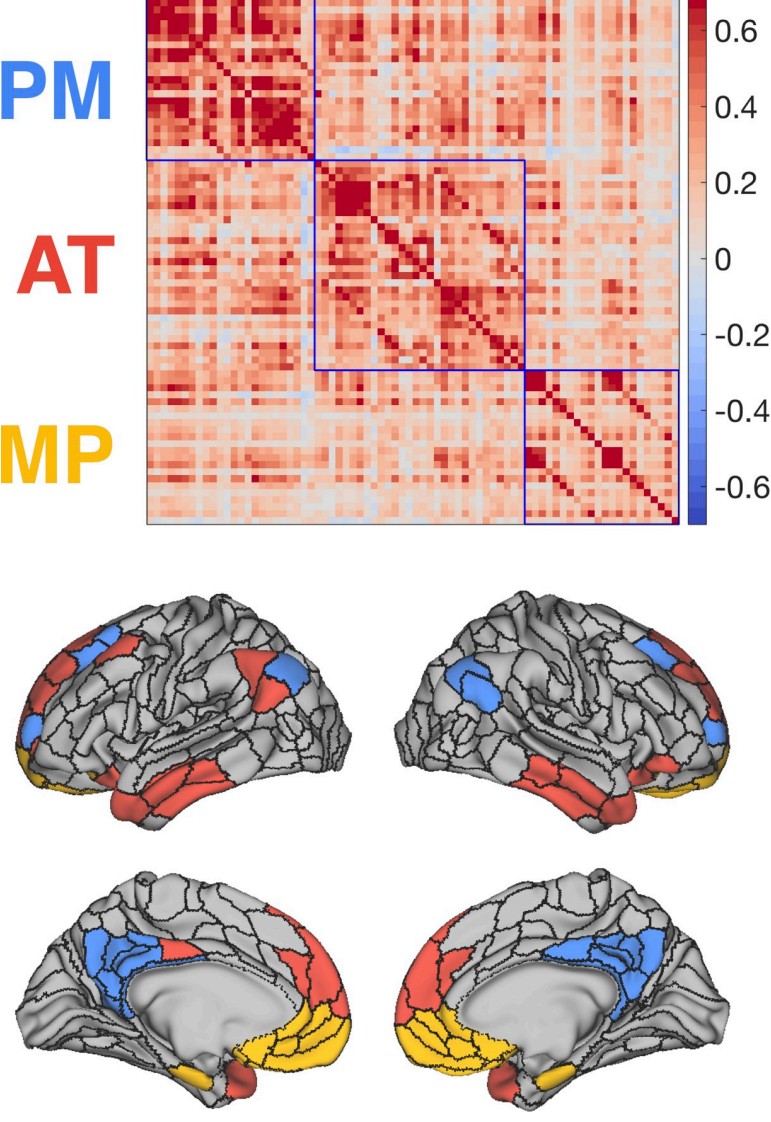

**Fig 3. The DMN can be partitioned into 3 interconnected subnetworks. (A)** FC matrix organized by subnetwork. Color bar represents Fisher Z-transformed correlation values. **(B)** Inflated cortical surface, colored according to community membership of subnetworks. AT, anterior temporal; DMN, default mode network; FC, functional connectivity; MP, medial prefrontal; PM, posterior medial. Data can be found at https://github.com/ajbarn/hippo_nets.

To visualize the topology of connections between these DMN subnetworks and the other large-scale networks defined earlier, we constructed a force-directed graph in abstract graph space (Fig 4). The ForceAtlas2 algorithm was used to construct the layout by modeling the graph as a physical system in which nodes repel each other and the FC between the nodes act as springs [54]. Regions within the PM subnetwork tightly clustered, and several nodes showed prominent cross-network connectivity with the MTN and frontoparietal network. The AT subnetwork showed cross-network interactions with the language and frontoparietal network, and the MP subnetwork was less clustered with fewer out of DMN connections.

To quantify the diversity of connections in the DMN subnetworks to the rest of the brain, we calculated participation coefficients—a graph theory metric that describes the extent to which a node's functional connections are spread across different communities. Nodes with a high participation coefficient are connected not only within their own community, but also communicate relatively more with nodes in other communities. This makes them well positioned to serve as regions that can integrate information from or coordinate activity toward multiple communities. Disruption of such nodes can lead to altered network function and cognitive impairment [62–65]. Here, we examined the diversity of connections for the nodes in each of the DMN subnetworks to examine how information may be assimilated into the DMN from communities across the brain and how information may be integrated within the DMN itself. Of the subnetworks, the AT subnetwork had the greatest diversity of connections to nodes outside of the DMN (AT > PM: $t(43) = 3$, $p = 0.004$; AT > MP: $t(32) = 2.7$, $p = 0.01$). As shown in Fig 4B, the PM subnetwork had several nodes with relatively high participation. These nodes connect to MTN, and the frontoparietal network, but the majority of the PM subnetwork showed less out of network connectivity. The MP subnetwork, on the other hand, had relatively low participation outside of the DMN.

We next addressed the extent to which nodes in the DMN showed diverse connectivity with nodes in the other DMN subnetworks. Interestingly, regions in the MP subnetwork exhibited the highest participation coefficients across the DMN subnetworks (MP > PM: $t(28) = 2.7$, $p = 0.01$; MP > AT: $t(35) = 3.7$, $p < 0.001$). These findings substantiate what can be seen in Fig 4A—that nodes in the MP subnetwork are positioned to integrate information across DMN subnetworks, whereas nodes in the AT subnetwork may interface with networks outside of the DMN.

## Cortico-hippocampal network connectivity

Having identified cortical networks and subnetworks that connect to the hippocampus (cortico-hippocampal networks), we then examined whether these networks showed a preference in connectivity along the hippocampal long axis. Previous research has shown differential FC along the long axis of the hippocampus [24,28,66,67], with the anterior hippocampus demonstrating greater connectivity to orbitofrontal cortex and temporal pole and the posterior hippocampus demonstrating greater connectivity to retrosplenial cortex and precuneus. We, therefore, hypothesized that the anterior hippocampus should have preferential connectivity to the AT subnetwork, whereas the posterior hippocampus would have preferred connectivity with the PM subnetwork. These predictions were partially confirmed by our findings. The anterior hippocampus showed stronger connectivity to the AT and MP subnetworks ($t(37) = 8.2$, $p < 0.00001$; $t(37) = 6.5$, $p < 0.00001$), and the posterior hippocampus showed stronger connectivity to the MTN than the anterior hippocampus ($t(37) = 2.6$, $p = 0.01$), but there was no significant difference in anterior and posterior hippocampal connectivity with the PM subnetwork ($t(37) = .41$, $p = 0.68$) (Fig 5A). To determine whether these differences were evenly spread across the networks or driven by particular regions within the networks, we contrasted

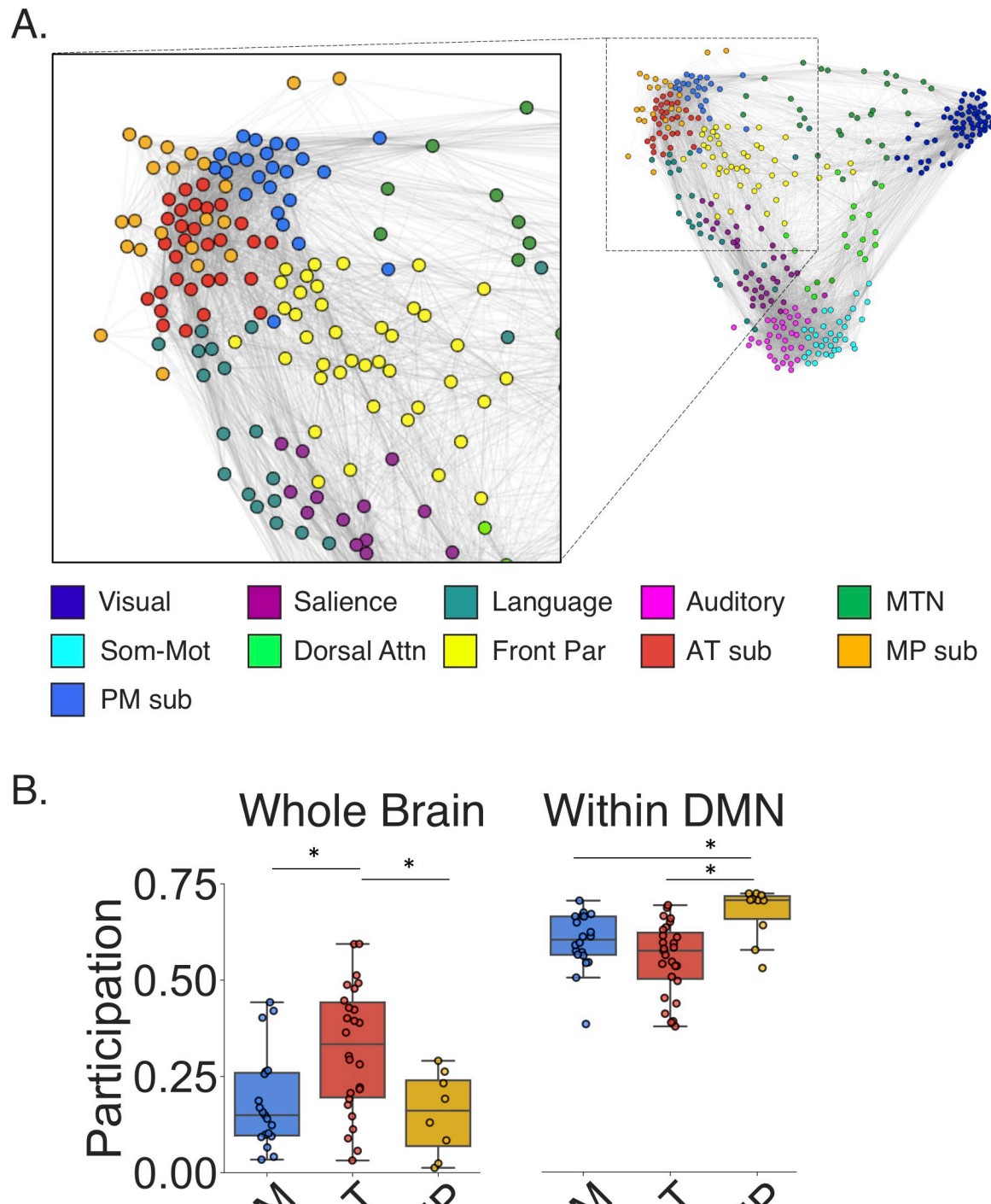

**Fig 4. DMN subnetworks interact with language, frontoparietal, and MTNs. (A)** Force-directed graph of the group-averaged networks with the DMN relabeled according to subnetwork membership. **(B)** Boxplots showing participation coefficient for DMN nodes to the rest of the brain (Whole Brain) and participation coefficient for DMN nodes to subnetworks of the DMN (Within DMN). AT, anterior temporal; Attn, attention; DMN, default mode network; Front Par, frontoparietal; MP, medial prefrontal; MTN, medial temporal network; PM, posterior medial; Som-Mot, somatomotor. * indicates significant difference at $p < 0.05$. Data can be found at https://github.com/ajbarn/hippo_nets.

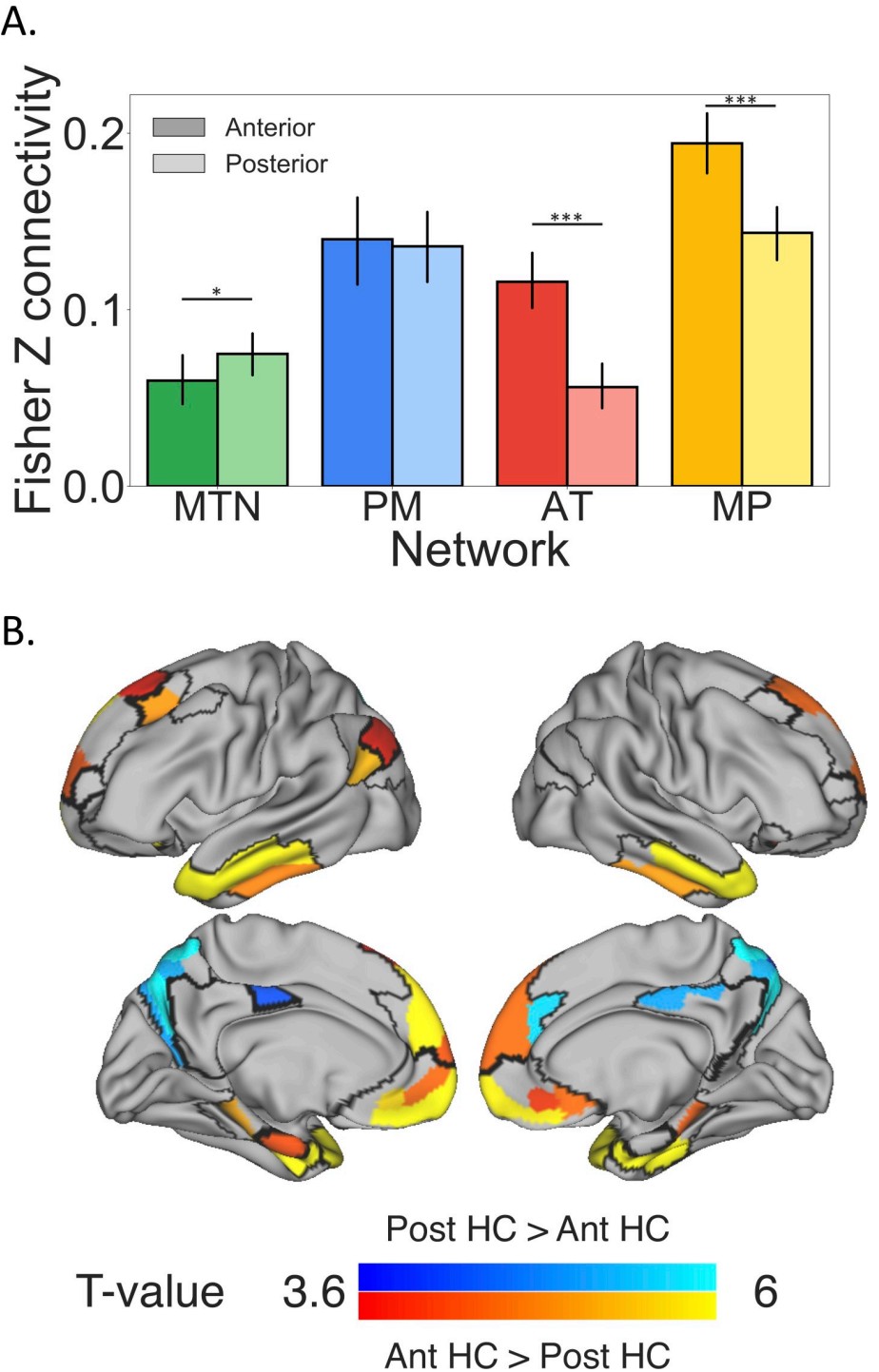

**Fig 5. Regions within the MTN and DMN subnetworks exhibit differential FC with the anterior and posterior hippocampus. (A)** Bar graphs of FC from each of the cortical networks to the anterior and posterior hippocampus. Error bars show 95% confidence interval. **(B)** T-contrast of anterior hippocampal connectivity versus posterior hippocampal connectivity, with warm colors showing regions that had stronger anterior hippocampal connectivity and cool colors showing regions that had stronger posterior hippocampal connectivity. Cortico-hippocampal network boundaries are outlined on the surface of the brain. These results indicate that the temporopolar and mPFC show preferential connectivity with the anterior hippocampus, whereas PM parietal cortex shows preferential connectivity with the posterior hippocampus. Ant, anterior; AT, anterior temporal; DMN, default mode network; FC, functional connectivity; HC, hippocampus; MP, medial prefrontal; mPFC, medial prefrontal cortex; MTN, medial temporal

network; PM, posterior medial; Post, posterior. * $p < 0.05$, *** $p < 0.001$. Data can be found at https://github.com/ajbarn/hippo_nets.

anterior and posterior hippocampal connectivity at a region-to-region level. As expected from the network-level analysis, we observed significantly greater anterior hippocampal connectivity to MP, AT, and also anterior medial temporal cortex, whereas the posterior hippocampus had significantly greater connectivity to the parietal occipital sulcus, precuneus, and dorsal PCC. Thus, the anterior–posterior differences are fairly consistent in the AT and MP subnetworks, but within the MTN, the overall anterior–posterior differences are driven by relatively higher posterior hippocampal connectivity in the medial parietal cortex (Fig 5B).

Based on neuropsychological evidence demonstrating distinct functional roles of the left and right hippocampus [68–70], we next investigated whether the left and right hippocampus differed in their connectivity to the cortical networks. We observed significant hemispheric laterality effects with the right hippocampus having greater connectivity to the MTN than the left ($t(37) = 3.8$, $p = 0.0001$) and the left hippocampus having greater connectivity to AT subnetwork ($t(37) = 2.6$, $p = 0.01$). There was no significant laterality effect in the PM ($t(37) = 0.29$, $p = 0.77$) and MP subnetworks ($t(37) = 0.98$, $p = 0.33$). This connectivity difference might help to explain differentiated function of the left and right hippocampus.

## Regions within the same community represent similar kinds of information during a memory task

We next sought to determine whether network membership was related to functional relevance of these subnetworks using an independent dataset in which participants performed a memory retrieval task (described in Mizrak and colleagues [71] and in Methods section) by examining the representational profile similarity between regions within the same network or in different networks [17,72]. Multivoxel pattern similarity was calculated between trials, which created a trial-by-trial representational similarity matrix for every ROI in our cortico-hippocampal networks. Here, we assume that trial-by-trial fluctuations in pattern similarity for a given ROI are driven by trial-by-trial fluctuations in features for which the ROI is sensitive. These fluctuations across trials represent the ROI's representational profile. We sought to determine whether regions within the same network had greater similarity in their representational profiles—and thus represent similar features—compared to regions outside their network.

Here, we calculated the similarity of representational profiles between ROIs by correlating each ROI's representational similarity matrix to every other ROI. We then averaged the representational profile similarities for pairs of ROIs within the same network to create a mean within-network similarity value, for each subject (we removed the similarity of each ROI with itself to avoid inflating the within-network similarity). To create a mean between-network similarity value, for each subject, we averaged together the representational profile similarity values for pairings of ROIs that were in different networks. For the cortico-hippocampal networks identified, we contrasted the mean within-network similarity with the mean between-network similarity using a repeated measures ANOVA and observed that regions within the same network had higher profile similarity compared to regions in different networks, $t(21) = 18.5$, $p < 0.001$ (Fig 6). This effect persists even when accounting for the similarity in trial-by-trial mean BOLD activity ($t(21) = 12.1$, $p < 0.001$), suggesting it is not simply a result of univariate activity similarity.

Another possible explanation of these findings is that regions in the same network tend to be closer in spatial proximity, and there is some spatial nonindependence of adjacent ROIs of

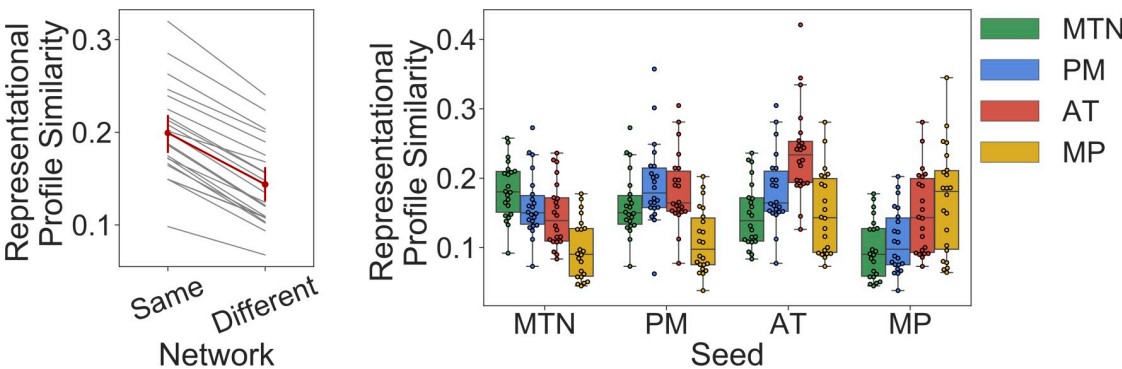

**Fig 6. Regions show higher representational profile similarity with community members compared to regions in other communities.** Left: individual subject representational profile similarity of regions within the same network or between network in gray lines with the group average in red, with 95% confidence intervals represented by error bars. Right: boxplot of average region-to-region representational profile similarity between each cortical-hippocampal network (labeled along the x-axis) to itself and every other cortico-hippocampal network (each represented by a colored bar) across the sample (see S6 Fig for the representational profile similarity of the cortico-hippocampal networks compared to every other cortical network). Individual participants represented as dots. AT, anterior temporal; MP, medial prefrontal; MTN, medial temporal network; PM, posterior medial. Data can be found at https://github.com/ajbarn/hippo_nets.

these representational effects. To account for this spatial proximity, we created an ROI-by-ROI distance matrix by calculating the Euclidean distance between the center of gravity of all pairs of ROIs. We then regressed out the influence of spatial proximity from the representational profile matrix using this distance matrix and repeated the analysis. After statistically removing the influence of spatial proximity, we still observed that ROIs within the same network had greater representational profile similarity compared to pairs of ROIs in different networks, $t(21) = 9.3$, $p < 0.001$.

## Neurosynth decoding

Having found evidence that regions within the same cortico-hippocampal networks carry similar information, we next ran a meta-analysis on Neurosynth [73]—a neuroimaging database with results from over 10,000 studies—to better understand the role these networks may play in cognition. Neurosynth enables meta-analyses of whole-brain activation maps associated with specific terms used in cognitive neuroimaging studies. We correlated meta-analytic activation maps of every term in the Neurosynth database with the binarized volumetric mask of each cortico-hippocampal network (i.e., 1 mask for each network comprised of every ROI in the network in MNI space). By examining the top cognitive terms associated with each network we observed that the term "autobiographical" was the only term that was among the top 20 terms identified in all 3 networks. Activation in the MTN and PM subnetwork was disproportionately associated with studies involving "episodic memory," whereas this term did not appear in the top 20 search terms for the AT and MP subnetworks (Fig 7). The MTN also showed strong overlap with meta-analytic maps for preference for navigation ($r = 0.15$) and scenes ($r = 0.12$), and the PM subnetwork showed strong overlap with self-referential ($r = 0.13$) and theory of mind terms ($r = 0.08$). The top terms associated with AT subnetwork related to theory of mind ($r = 0.22$), intentions ($r = 0.17$), mental states ($r = 0.22$), and social cognition ($r = 0.19$), while the top terms associated with MP subnetwork were value ($r = 0.17$), fear ($r = 0.16$), emotion ($r = 0.16$), and terms related to emotional valence ($r = 0.14$). Weightings for each network are available in S2 Table.

### Anterior Temporal

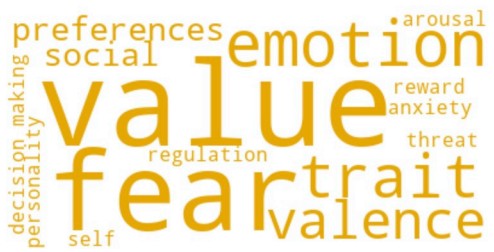

### Posterior Medial

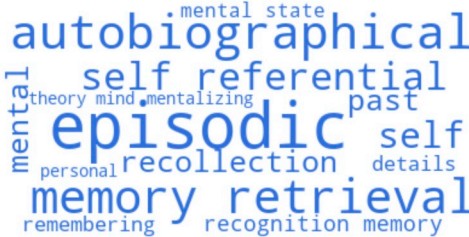

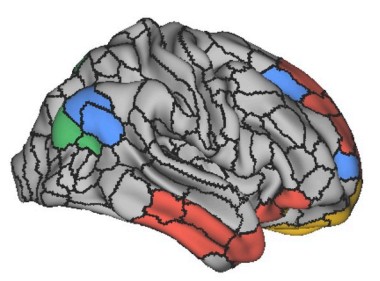

### Medial Prefrontal

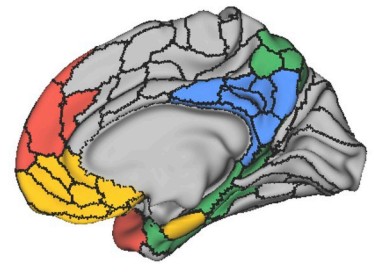

### Medial Temporal

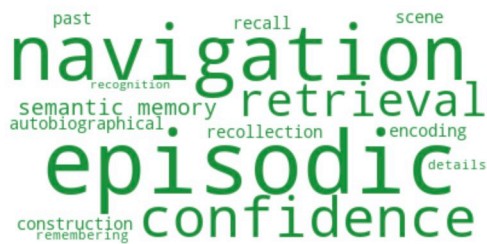

**Fig 7. Terms associated with activation of each cortico-hippocampal network.** Word clouds are colored according to the network they describe which are displayed on the inflated brain. The size of the terms in the word cloud relates to how strongly the meta-analytic activity of that term correlates with the spatial extent of the network. Red, AT subnetwork; Blue, PM subnetwork; Yellow, MP subnetwork; Green, medial temporal network. Data can be found at https://github.com/ajbarn/hippo_nets.

## Discussion

In the present study, we used data-driven analyses of resting-state fMRI to identify and comprehensively characterize cortico-hippocampal network connectivity. Analyses showed that the MTN, a cluster of regions in the medial temporal and dorsomedial parietal lobes, could be differentiated from the DMN, and we replicated this finding in a secondary dataset. We also found that the DMN could be meaningfully subdivided into 3 subnetworks: the PM subnetwork, encompassing the posterior cingulate, retrosplenial, lateral parietal, and dorsal lateral prefrontal cortex; the AT subnetwork, encompassing the temporopolar, lateral orbitofrontal, and dorsal mPFC; and the MP subnetwork, encompassing the ventral MP and entorhinal cortex. The MTN and DMN subnetworks could be differentiated by connectivity strength along the long axis of the hippocampus and in information carried by multivoxel activity patterns in an independent dataset.

Prior work on the neurobiology of memory has drawn on the idea that the hippocampal formation primarily interacts with the parahippocampal cortex and perirhinal cortex, such that these areas collectively comprise an "MTL memory system" [2] that is functionally distinct from surrounding cortical areas. More recently, we and others proposed that parahippocampal cortex and perirhinal cortex are embedded in larger scale cortico-hippocampal networks as evidenced by divergent anatomical pathways found in rodents and nonhuman primates [19–22] and by differing FC of the parahippocampal cortex and perirhinal cortex in humans [24,25,74]. In this "PM/AT" framework, the parahippocampal cortex is a core region in the PM network, which also includes retrosplenial, posterior cingulate, and lateral parietal cortex, whereas the perirhinal cortex is a core region in the AT network, which also includes temporopolar and orbitofrontal cortex [29]. As shown in Fig 8, the present results were not fully

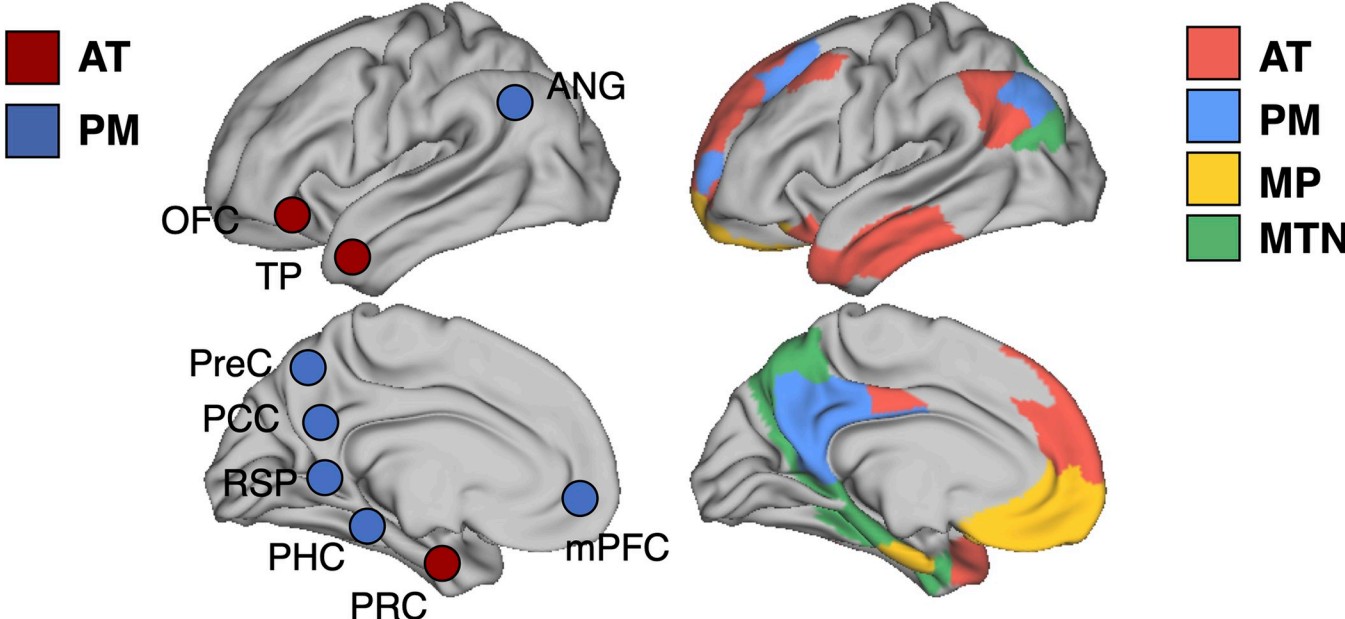

**Fig 8.** Left: hypothesized PM/AT divisions with PM nodes as blue and AT nodes as red [29,78]. Right: data-driven network assignments. ANG, angular gyrus; AT, anterior temporal; mPFC, medial prefrontal cortex; OFC, orbitofrontal cortex; PCC, posterior cingulate cortex; PHC, parahippocampal cortex, PM, posterior medial; PRC, perirhinal cortex; PreC, precuneus; RSP, retrosplenial cortex; TP, temporal pole.

consistent with either the original PM/AT or the MTL memory systems frameworks. We found that the perirhinal cortex, parahippocampal cortex, parieto-occipital sulcus, inferior lateral parietal, and dorsal medial parietal formed the MTN—a set of regions previously described by Kravitz and colleagues [75] as the "parieto–medial temporal pathway". Our results converge with some previous data-driven community detection studies [15,42,49,76], showing that the MTN can be differentiated from the DMN (but see [45,77]). Critically, both the MTN and DMN showed strong FC with the hippocampus.

Although the present work shows that the MTN could be differentiated from the DMN, they do not imply that regions within the MTN are homogenous in terms of connectivity or function. Prior anatomical studies in animal models [79,80] and fMRI studies in humans have shown that the perirhinal cortex and parahippocampal cortex have different FC patterns, particularly when contrasted against each other [24–26,81]. However, it is also clear that the 2 regions frequently show strong FC to each other and considerable overlap in the spatial extent of their FC [24,25,81]. For example, Wang and colleagues [25] and Zhuo and colleagues [81] showed that the parahippocampal cortex and posterior perirhinal cortex show high connectivity with each other, and with medial parietal, MP, lateral parietal, and temporopolar regions. In line with this evidence, the results presented here demonstrate that, after accounting for the topology of functional connections across the entire neocortex, the MTN can be segregated from the DMN.

Treating the brain as an abstract graph, we observed that the MTN serves as a critical bridge between the visual network, the DMN, and the hippocampus, such that removal of the MTN results in disrupted information flow between the visual network and DMN/hippocampus. Damage to perirhinal and parahippocampal cortex is often associated with a significant amnesic syndrome characterized by both episodic memory deficits and impaired acquisition of new semantic knowledge [11,82,83]. Our results are consistent with the idea that amnesia caused by damage to these MTL cortical regions may be due to a disconnection syndrome [84,85].

Specifically, damage to these areas should impair the ability to learn links between information about the external visual world carried by the visual network and abstract knowledge about the latent structure of events and individuals carried by the DMN [78,86]. Results consistent with this idea were reported in a recent study that used a novel analytic method to map the FC of regions in the brain where lesion damage resulted in amnesia [11]. The FC network that was common to all the lesions encompassed the DMN, MTN, and encroached into visual network regions as described in this study, underscoring the importance of these networks in memory processing. Another study of amnesia following traumatic brain injury observed reduced FC between the parahippocampal cortex and PCC was related to memory impairment and normalized when memory functioning improved [87].

The distinction between the MTN and the DMN provides insight into recent studies that suggested heterogeneity within the PM network [14,38,52,88]. Based on its network properties, the MTN sits in a reasonable position to represent higher-order features of the visual world to be encoded by the hippocampus. Supporting this interpretation, MTL regions have been implicated not only in episodic memory, but also perception [57] and memory-guided eye movements [89]. The MTN also has strong connectivity with the DMN, which is isolated from networks involved in primary sensation, and, thus, positioned to represent more abstract information, such as knowledge about different types of events [90–92]. Recent theories of event cognition have suggested a separation between content (higher-order representations of current environment) and structure (abstracted knowledge pertaining the typical way in which sequences of events unfold) [86,93]. It is possible that the MTN represents content of events and feeds that content to the DMN that represents the abstracted event structure. Indeed, recent evidence has shown that regions in the MTN are more active during recall focused on perceptual features [39] and during encoding of events in the absence of a strong semantic framework [40], whereas regions in the DMN are more active during recall focused on conceptual or thematic elements [39] and during encoding of events that can be interpreted in the context of prior knowledge [40]. Further, multivariate patterns in DMN regions can classify broad event types like restaurant versus airport events, suggesting that they have represent abstract event structure [91].

Although these studies support a distinction between the MTN and DMN, it is important to note that these networks likely work in tandem to support retrieval of both content and structure, such as during recall of naturalistic events spanning minutes [94]. Rather than viewing these systems as completely independent, a more measured interpretation is that task demands may determine what networks are recruited for retrieval. For example, one might expect that, even within the MTN, a contrast that focuses on the distinction between different types of content, such as scenes versus objects, may split regions within the MTN that have a bias toward those content types [35]. These findings demonstrate the necessity of our intrinsic connectivity analyses to clarify that these differences in patterns of functional activity are likely due to the presence of distinct functional networks.

The second major finding in the present study is that the DMN could be differentiated into 3 subnetworks: the PM, AT, and MP subnetworks. The PM subnetwork consisted of posterior cingulate, retrosplenial cortex, angular gyrus, and dorsal prefrontal cortex, whereas the AT subnetwork consisted of temporopolar, lateral temporal, orbitofrontal, and dorsal mPFC. The collection of regions in the PM and AT subnetwork is in line with the PM/AT framework [24,25], with the notable exception that the parahippocampal cortex and precuneus were not in the PM subnetwork and the perirhinal cortex was not in the AT subnetwork. The PM and AT subnetworks also resemble DMN subnetworks identified by previous data-driven methods [15,41,42], confirming that the PM and AT distinctions theorized do indeed persist, but they form distinct communities from their MTN counterparts when examining their connectivity

architecture. Andrews-Hanna and colleagues [41] also fractionated the DMN, identifying an MTL subnetwork that resembles our MTN, a dorsal medial subnetwork that resembles what we have termed the "AT network" and a "core midline" network comprising both PCC and mPFC. This midline network differs from our partition which separates the mPFC and PCC, but it is important to note that the core midline network described in Andrews-Hanna and colleagues [41] was created based on high betweenness centrality (a graph theory metric thought to be related to "hubness") of the mPFC and PCC, rather than from a data-driven partitioning procedure.

The MP subnetwork identified consists of mPFC, along with the entorhinal cortex, and was not explicitly predicted by prior partitions [15,41,42]. However, recent work that has pushed the resolution parameters of community detection have observed the mPFC forming a separate network from the PM cortex [95]. The mPFC has long been distinguished from the orbitofrontal cortex based on anatomical connectivity [21,22,96] and is often grouped together with the PCC as one of the "core hubs" of the DMN [15,41,42]. The entorhinal cortex certainly has dense interconnections with the mPFC [96], but also is strongly interconnected with the hippocampus and adjacent medial temporal cortex [19,22,97]. Interestingly, the entorhinal cortex often is part of the DMN even when the MTL cortex is distinguished from the rest of the DMN ([76,98], but see [15,49]). Furthermore, network analysis on histologically defined axonal connections between cortical regions in rodents showed that the lateral entorhinal cortex was part of a community with the prefrontal cortex, consistent with our partition, although the medial entorhinal cortex was placed in a community with the hippocampus, retrosplenial cortex, and subiculum [99]. A distinction in the FC of entorhinal cortex subregions has been shown in high-resolution human resting-state data [23,37], but the spatial resolution of the current study is unable to resolve these subregions, and future work will be important to further characterize its connectivity in humans.

We examined the topology of how these subnetworks connect to each other and the rest of the brain finding that the MP subnetwork had the highest within-DMN participation coefficient, with nodes connecting broadly to AT and PM regions. Conversely, the MP subnetwork had a low out of DMN participation coefficient, especially relative to the AT subnetwork. Our results provide converging evidence for a recent study published by Gordon and colleagues [95] that also found that a set of lateral networks resembling the AT subnetwork here showed the highest participation coefficient out of network. Their AT equivalent showed intercommunication with language and frontoparietal networks, as was also seen here. The participation coefficient findings suggest that the AT subnetwork and several high participation PM nodes are positioned to mediate information transfer in and out of the DMN, whereas the MP subnetwork may integrate and coordinate information within the DMN. A recent review of the human lesion literature has reported that robust memory impairment can be observed following mPFC damage and suggested that this may be due to the fact that the mPFC has a role in initiating and coordinating cognitive processes for retrieval and episodic simulation [100]. Given the network findings here, and implications from the lesion literature, the MP subnetwork is well positioned for higher-order integration and coordination of memory-guided activity.

Differentiation of the 3 DMN subnetworks was reinforced by evidence from an independent dataset in which we found that regions grouped in the same network represented similar kinds of information, as compared to regions in different networks. These findings replicate and extend results from previous studies that identified different functional differences between different cortico-hippocampal networks [17,72]. For instance, Ritchey and colleagues [17] used community detection to differentiate networks of regions hypothesized to show different patterns of MTL connectivity. Ritchey and colleagues [17] interrogated these networks

in an independent study of episodic memory retrieval, as did Inhoff and colleagues [72] during a concept learning task. The functional distinction of these networks was supported by findings showing that regions grouped in the same network had greater representational profile similarity [17] and activation profile similarity than regions grouped in different networks during learning and memory tasks [17,72]. In these past studies, the regions used in network construction were those that showed a significant difference in FC between the perirhinal cortex and parahippocampal cortex, being motivated by anatomical evidence suggesting parallel pathways in the MTL [19].

Whereas the seed selection that defined the networks in Ritchey and colleagues [17] and Inhoff and colleagues [72] was based specifically on regions that had different FC between parahippocampal cortex and perirhinal cortex, in the current study, we did not impose any preexisting assumptions and identified networks holistically in the context of the entire brain. In the independent task dataset we examined, Mizrak and colleagues [71] had participants learn about desirability of foods in different stores where a latent task structure was available. Specifically, participants learned about customer preferences for 8 food items in 4 different grocery stores. The task was structured such that a food's desirability could be context dependent or context invariant. Some foods were desirable in some store contexts and not in others (context dependent); some foods were always desirable or undesirable no matter what store context they were paired with (context invariant). After learning the customer preferences in each store, participants were scanned while they decided whether a food was desired or not by customers based on previous learning for each food in each store context. During this memory-guided decision-making task, we saw reliably higher representational profile similarity for regions in the same network—a pattern consistently found in every subject highlighting the functional significance of these cortico-hippocampal networks and validating our delineation of these cortico-hippocampal networks. Mizrak and colleagues focused on hippocampus and OFC, examining how these 2 regions represented key information from the task. They showed that hippocampal and OFC representations differentiated context-dependent and context-invariant task states. The OFC and hippocampus have long been studied in the context of decision-making, but only more recently have these regions been seriously explored in conjunction in memory-guided decision-making [101,102]. Our findings of strong FC between the hippocampus and orbitofrontal regions, along with the joint representation of task structure in these regions found by Mizrak and colleagues, bolsters the idea that these regions are working in a coordinated fashion.

We also found that the MTN and DMN subnetworks differ in their connectivity across the anterior and posterior hippocampus. The hippocampus is known to vary along its long axis in terms of intrinsic connectivity, extrinsic connectivity, receptor distribution, and gene expression [19,20,103–106]. In humans, resting-state FC has been the primary tool for examining differences in connectivity along the hippocampal longitudinal axis, although distinctions between the anterior and posterior hippocampus have been observed using DTI [28]. The anterior hippocampus has been shown to have greater connectivity to inferior temporal, temporopolar, orbitofrontal, MP, and perirhinal cortex, whereas the posterior hippocampus has been shown to have greater connectivity to parahippocampal cortex, retrosplenial cortex, inferior parietal cortex, and precuneus, and anterior thalamus [24,26–28,66]. Here, we replicated and extended these findings to characterize hippocampal connectivity to the networks formed by these cortical regions. We found that the anterior hippocampus has relatively greater connectivity to the MP and AT subnetworks, whereas the posterior hippocampus has relatively greater connectivity to the MTN. The preferential connectivity to the anterior hippocampus was relatively consistent across all regions in the MP and AT subnetworks, but there was heterogeneity in the MTN's preference for the anterior and posterior hippocampus with the

anterior hippocampus having stronger connectivity to anterior MTL cortex and the posterior hippocampus having relatively stronger connectivity to dorsal medial parietal cortex. Theories regarding the functional differences between the anterior and posterior hippocampus are often informed by preferential long axis connectivity to the rest of the brain [16,107], but they do not consider how the rest of the brain connects to form networks. The results presented here suggest the possibility that functional differences between the anterior and posterior hippocampus may reflect differences in their connections to the subnetworks of the DMN.

To better understand the functional specializations of these networks, we used meta-analytic maps from the Neurosynth database to identify the cognitive constructs that have been reliably associated with regions in the DMN subnetworks. Activity within the MTN was associated with a set of terms heavily rooted in episodic memory and navigation. Activation in the MP subnetwork, in contrast, was highly associated with terms pertaining to emotion and value. As noted above, the MTN (in particular the dorsal precuneus and dorsal posterior cingulate) had greater connectivity to the posterior relative to the anterior hippocampus, whereas the MP subnetwork had greater connectivity with the anterior compared to posterior hippocampus. These connectivity patterns and neural decoding results converge with theories linking the posterior hippocampus to cognitive and navigational processes and the anterior hippocampus in motivation and emotional behavior [108]. We also found that activation in the AT subnetwork was associated with terms that relate to social cognition, theory of mind, and beliefs, in line with contemporary theories [109] and lesion evidence [110] of this network. In the PM subnetwork, Neurosynth decoding revealed cognitive terms related to episodic memory and the self. This is echoed by recent work by DiNicola and colleagues [111]. They identified subject-specific default mode subnetworks that corresponded to our PM and AT subnetworks using resting-state fMRI. They further observed that the AT-like subnetwork showed increased activation during a theory of mind task, whereas the PM-like subnetwork showed increased activation during an episodic projection task. It should be emphasized, however, that despite the fact that these networks can be differentiated, they are likely to work in tandem depending on task demands. For example, the Neurosynth analysis showed that the term "autobiographical" was strongly associated with all 4 cortico-hippocampal networks. This broad association may be due to the fact that autobiographical memory tasks usually involve retrieval of episodically rich events that dynamically unfold involving social interactions, and relevance to the self [112], touching on many of the purported functions of the identified networks.

Although the Neurosynth decoding analysis is exploratory, and by definition, post hoc, it can guide the generation of hypotheses to be tested in future studies. Further studies, similar to the work of DiNicola and colleagues [111], should examine whether MTN and DMN subnetworks segregate based on spatial, valuation, and social processes, within the same group of individuals. For example, during encoding and recall of complex events, we might expect regions within the MTN to represent high-level object and context features, regions within the AT subnetwork to represent interpretations of theory of mind and social interactions, and regions within the MP subnetwork to represent the emotional valence or perceived value of the event. Indeed, a recent preprint demonstrated a dissociation between the MTN and MP subnetwork during event simulation, with increased activity in the MTN occurring when subjects were instructed to simulate past or future events using cues that were selected to elicit high levels of vividness, whereas increased MP activity was observed when cues elicited simulations with higher emotional valence [113]. Further, based on the FC of the MP subnetwork, we hypothesize that it would be well suited to serve a coordinating role in network rearrangement to accomplish task goals and demands. Again, a recent study by Nawa and Ando [114] found that when elaborating on autobiographical memories, the ventromedial prefrontal cortex (vmPFC)

activity drove hippocampal activity, and this effect was augmented for more emotionally arousing memories. Further, damage to the mPFC is associated with difficulties in elaborating on memories unfold over time and events [100]. We hypothesize that these impairments may be associated with dysfunction of network dynamics due to a loss of the coordination normally provided by the MP subnetwork.

Finally, we note that recent studies have demonstrated the value in taking subject-specific approaches to delineating functional networks in a voxel-wise fashion [14,49]. This work has shown that there are individual differences in the location of boundaries between functional networks, although it is notable that the majority of the cortex is labeled consistently in the majority of participants [76]. A key goal of the present study was to provide a clear framework to guide analyses in future task-fMRI studies, in which it might not be feasible to obtain independent, subject-specific parcellations. We have listed each region and its community label in S3 Table, and the HCP-MMP atlas is available online (https://balsa.wustl.edu/study/show/RVVG [46]). Task fMRI studies that use group-level analyses can use the group-level characterization of cortico-hippocampal networks reported here in order to rigorously test hypotheses about the functions of these networks and how they may be implicated in memory disorders, as resting connectivity has shown to predict spread of pathology in Alzheimer disease [115], and DMN subnetwork perturbation has been demonstrated in MTL amnesia [116].

## Conclusions

The hippocampus affiliates with a broad set of regions that enable episodic retrieval. Here, we have shown that the MTN and 3 subnetworks of the DMN can be differentiated on the basis of their whole-brain FC. The 3 DMN subnetworks vary in connectivity along the hippocampal long axis, have distinct representational roles during a memory task, and topologically are connected by a set of hubs in the MP subnetwork. This subnetwork organization offers a novel framework to investigate event cognition and memory retrieval.

## Methods

### Participants

For the primary resting-state fMRI dataset, 45 healthy, young adult participants were recruited from the University of California, Davis and surrounding area ($N_{Females}$ = 26, mean age = 25.6 years [SD = 4.2 years]). All participants were right-handed and neurologically healthy. The study was approved by the Institutional Review Board of the University of California at Davis (IRB #637028) and adheres to all principles of the Belmont Report. All participants provided written informed consent prior to participation. Participants were compensated $20/hour for their time. While there is no specific effect size that can be taken to approximate power for identifying network communities, the current sample size is comparable to the cohort sample sizes from the seminal Power and colleagues [117] study investigating functional brain organization, but has over double the amount of sampled resting-state time points (25 minutes of rest as noted below). This amount of individual data has also been shown to be reliable in estimating network communities, showing 77% to 83% similarity (measured with Dice coefficient) on a subject-level basis [76,118].

To assess the generalizability of our findings, we accessed a secondary resting-state fMRI dataset consisting of 120 healthy young adults [119]. From this sample, we removed all participants who completed only one of the 2 scanning sessions, resulting in 76 participants to be preprocessed and analyzed. An additional 7 participants were excluded due to data quality (described below) resulting in 69 participants ($N_{females}$ = 31, mean age = 24.5 years, [SD = 2.5 years]). All participants reported no history of neurological or psychiatric disorders. This

dataset is available at https://openneuro.org/datasets/ds000243/versions/00001 and has been previously described in [120].

## MRI acquisition

MRI scanning for the primary dataset was performed using a 3T Siemens Skyra scanner system (Siemens, USA) with a 32-channel head coil. A T1-weighted structural images was acquired using a magnetization prepared rapid acquisition gradient echo pulse sequence (TR = 2,100 ms; TE = 2.98 ms; field of view = 256 mm$^2$; flip angle = 7˚; image matrix = 256 × 256, 208 axial slices with 1.0 mm$^3$ thickness; GRAPPA acceleration factor 2 with 24 reference lines). Functional images were acquired using a gradient echo-planar imaging (EPI) sequence (TR = 1,220 ms; TE = 24 ms; field of view = 192 mm$^2$; image matrix = 64 × 64; flip angle = 67˚; bandwidth = 2442 Hx/pixel; multiband factor = 2; 38 interleaved axial slices, voxel size = 3 mm$^3$ isotropic). Five runs of 5 minutes in duration were acquired for rest scans for a total of 25 minutes of resting-state fMRI data per subject. Participants were instructed to lay as still as possible and try not to fall asleep. Scanning parameters for the secondary dataset can be found in [120]. Each participant in the secondary dataset underwent an average of 12.8 minutes of resting-state fMRI (SD = 3.7 minutes).

## Anatomical preprocessing

T1-weighted anatomical scans were preprocessed using FreeSurfer, which included intensity normalization, removal of non-brain tissue, transformation to Talaraich space, and segmentation of gray matter, white matter, and cerebrospinal fluid (CSF). Surfaces were calculated for the white matter–gray matter and gray matter–pial interface. Surface-based registration to the HCP-MMP1.0 atlas [46] was performed, and subject-specific cortical regions were calculated according to atlas boundaries. Surface-based cortical regions were converted to volumetric regions of interest and transformed into functional native space. The hippocampus was segmented in FreeSurfer in an automated fashion. Manual adjustments were done to correct misclassified voxels, and the hippocampus was divided into anterior and posterior segments based off of the disappearance of the uncal apex [16], with the posterior hippocampus designated as all of the hippocampus posterior to the disappearance of the uncal apex on a coronal slice.

## fMRI preprocessing

Both the primary and secondary resting-state fMRI datasets underwent the same preprocessing as described below. Functional preprocessing was performed using *fMRIPrep* 1.4.1 ([121,122], RRID:SCR_016216), which is based on *Nipype* 1.2.0 ([123,124], RRID: SCR_002502). For each of the 5 BOLD runs found per subject (across all tasks and sessions), the following preprocessing was performed. First, a reference volume and its skull-stripped version were generated using a custom methodology of *fMRIPrep*. A deformation field to correct for susceptibility distortions was estimated based on 2 EPI references with opposing phase-encoding directions, using 3dQwarp from Cox and Hyde [125] (AFNI 20160207). Based on the estimated susceptibility distortion, an unwarped BOLD reference was calculated for a more accurate co-registration with the anatomical reference. The BOLD reference was then co-registered to the T1w reference using flirt (FSL 5.0.9 [126]) with the boundary-based registration [127] cost function. Co-registration was configured with nine degrees of freedom to account for distortions remaining in the BOLD reference. Head motion parameters with respect to the BOLD reference (transformation matrices and 6 corresponding rotation and translation parameters) are estimated before any spatiotemporal filtering using mcflirt (FSL 5.0.9 [128]). The BOLD time series were resampled onto their original, native space by

applying a single, composite transform to correct for head motion and susceptibility distortions. These resampled BOLD time series will be referred to as preprocessed BOLD. Several confounding time series were calculated based on the preprocessed BOLD as part of *fMRIPrep*; however, only the head motion estimates calculated in the correction step and frame displacement (FD) were used in subsequent analyses. All resamplings can be performed with a single interpolation step by composing all the pertinent transformations (i.e., head motion transform matrices, susceptibility distortion correction when available, and co-registrations to anatomical space). Gridded (volumetric) resamplings were performed using antsApplyTransforms (ANTs), configured with Lanczos interpolation to minimize the smoothing effects of other kernels [129]. Many internal operations of *fMRIPrep* use *Nilearn* 0.5.2 ([130], RRID: SCR_001362), mostly within the functional processing workflow. For more details of the pipeline, see https://fmriprep.readthedocs.io/en/stable/workflows.html.

The preprocessed BOLD time series, anatomical images, and native space Glasser parcels were imported into CONN Toolbox version 18b (www.nitrc.org/projects/conn, RRID: SCR_009550). Based on Ciric and colleagues' [131] assessment of 14 common denoising protocols, we selected the "9p" protocol because it was shown to facilitate the highest functional network identifiability (see [131]; Fig 5B). BOLD time series were demeaned, linear and quadratic trends were removed, and bandpass filtered between 0.008 and 0.09 Hz. The 9p protocol includes as confound regressors 6 motion parameters, white matter, CSF, and global signal regression. All regressors were bandpass filtered to maintain the same frequency range as the data. Visual inspection of the frequency distributions of FC values as well as quality control plots of the correlation between mean FD and FC values was performed to identify any aberrant runs or subjects. Five outlier subjects were identified in the primary dataset, and 4 outlier subjects were identified in the secondary dataset, which each corresponded to previously identified subjects having a mean FD value of greater than 0.15 mm. These subjects were excluded from further analyses. A further 3 subjects were excluded from the secondary dataset due to anomalous FC distributions after denoising. FC matrices were created for each subject by computing Pearson correlations between all possible pairs of each region's confound-corrected time series. Finally, each correlation value was Fisher z-transformed with the inverse hyperbolic tangent function.

## Community detection

Cortical FC matrices from all subjects were thresholded to exclude negative connections that may be introduced by global signal regression [132,133] and then averaged together. Using the resulting group-averaged connectivity matrix, community detection was performed using the Louvain method [47] via the brain connectivity toolbox (https://sites.google.com/site/bctnet/), which iteratively performs a greedy optimization of modularity by randomly selecting nodes and merging them into the community that maximally increases modularity, until no more gains in modularity are possible. Modularity (Q) describes how well a network can be divided into communities that have higher within-community connectivity than would be expected by chance and can range from −1 to 1, with negative values indicating fewer intracommunity connections than would be expected by chance and positive values indicating higher intracommunity connections than would be expected by chance [59]. We used a method of the Louvain community detection algorithm adapted to accommodate weighted graphs [134], which accepts the weighted, group average FC matrix. This algorithm can be tuned using a resolution parameter, gamma, that biases the algorithm toward producing few, large networks (low gamma) or toward many, smaller networks (high gamma). To determine the resolution parameter in a principled manner, we adopted the approach used by Ji and colleagues [45].

The criteria included (i) separation of the primary sensory–motor networks (visual, auditory, and somatomotor) from all other networks; (ii) high stability across nearby parameters (similar network partitions across neighboring parameter settings); and (iii) optimized modularity.

Given that the Louvain algorithm detection is influenced by a random starting point, for every tested resolution parameter, we ran the algorithm 1,000 times. To determine the optimal partitioning solution at each resolution, gamma, we examined how consistent a given partitioning solution was to every other solution produced over the 1,000 iterations at the same resolution. Consistency was calculated using the z-Rand score [48] (http://commdetect.weebly.com/). For each partition solution, we weighted the mean z-Rand score (consistency) of the partition with the modularity value of that partition to select a partition solution that was both highly consistent at a given resolution and had high modularity. These methods are in keeping with Ji and colleagues [45]. Using the default parameter, gamma = 1, we could not satisfy criteria (i) as the somatomotor and auditory network were grouped together (also seen in Ji and colleagues [45]). Thus, we ran a parameter sweep from gamma 1 to 2.8 by increments of 0.005 until we identified a resolution parameter at which the somatomotor and auditory network separated consistently and produced the highest modularity-weighted z-Rand score.

To assess the internal validity of our community partition, we performed a split-half analysis in which our primary sample was split into 2 equal halves, matched for gender (Group 1, $N_{Female}$ = 13; Group 2, $N_{Female}$ = 13) and then age (Group 1, mean age = 25 years [SD = 4.3 years]; Group 2, mean age = 26 years [SD = 4.1 years]). Community detection analyses and criteria described above were applied in both halves separately and compared using the z-Rand index.

To assess the external validity of our community partition, we acquired a secondary dataset, from OpenNeuro (https://openneuro.org/datasets/ds000243/versions/00001) and repeated the community detection procedure described above. To compare the primary and secondary datasets, we selected the resolution parameter that produced the same number of communities in the secondary dataset that were observed in the primary dataset. The resulting partition was compared to the primary dataset using z-Rand index.

## tSNR comparison

Due to observations that one network in the secondary dataset was comprised of low FC regions of interest in the secondary dataset, we compared the tSNRs between our primary dataset and secondary dataset. In each dataset, for each subject, tSNR was calculated by first temporally concatenating all functional runs. Next, we calculated the temporal mean signal and temporal standard deviation at each voxel. Finally, we divided the temporal mean signal by the temporal standard deviation at each voxel. We used FSL command line functions to complete these calculations. For each subject, we masked each region of interest in the tSNR image and calculated the mean tSNR of all voxels within each region of interest.

## Path length analysis

Using the brain connectivity toolbox (https://sites.google.com/site/bctnet/), weighted path length was calculated between all nodes in the brain for each subject, after excluding negative connectivity weights. Path length is the fewest number of links that need to be traversed to connect 2 nodes. To calculate the weighted path length, the FC weights were inverted creating a distance matrix in which nodes with high FC had low distance. Then, the shortest path between each pair of nodes was calculated providing a value that represented the minimal the total weighted distance that needed to be traversed to connect 2 nodes.

We hypothesized that removal of the MTN would reduce the ability of the visual network and DMN/hippocampus to connect with each other. To test this hypothesis, all the nodes from the MTN were removed from each subject's network, and the average weighted path length was calculated between visual network and DMN and between visual network and hippocampus. Because removal of nodes can only lead to an increase in path length (shorter paths can be removed but will never be added following removal of nodes), we compared the weighted path length following removal of the MTN to the weighted path length following removal of every other network, using paired $t$ tests. As a secondary control, we also compared the weighted path length following removal of the MTN to the distribution of weighted path lengths following removal of 30 random nodes (other than the DMN, visual network and hippocampus) over 1,000 iterations. For this, we calculated the z-score of the weighted path length following removal of the MTN compared to the distribution of weighted path lengths following removal of 30 nodes over 1,000 iterations. This was done for each subject such that each subject had a corresponding z-score. A 1-sample $t$ test was then performed on the z-score values to examine whether they were significantly greater than 0.

We also ran analyses to control for spatial proximity. Path length results following removal of the MTN may be driven by the MTN's spatial proximity to the visual network as well as the DMN/hippocampus. Thus, we identified a set of 32 non-MTN regions that was matched for distance to both the visual and DMN compared to the MTN. We also were able to identify 31 non-MTN regions that were matched for spatial distance to both the visual network and hippocampus compared to the MTN. To do so, we calculated the region-to-region Euclidean distance for each pair of ROIs for each subject. We then calculated the average distance between each MTN ROI to each DMN ROI, the average distance between each MTN ROI and each VIS ROI, and the average distance between each MTN ROI and hippocampal ROI, separately for each hemisphere. Then, we identified 2 sets of control ROIs, one such that the average distance between this control set and the DMN and the control set and the visual network would be within ±1 SD from the mean MTN-DMN and MTN-visual network distance and the other control set was the same except instead of controlling for MTN-DMN distance it controlled for MTN-hippocampal distance. Having found these sets that fulfilled these criteria, we deleted the sets and recalculated path length between the visual network and DMN and visual network and hippocampus and compared the resulting path lengths to deletion of every other network using paired $t$ tests.

## Community partitioning of the default mode network

From the identified whole-brain partitioning, we selected the DMN based on visual similarity to previously identified DMN solutions [42,45,135]. We repeated the community detection procedure described above on the identified DMN (1,000 iterations of Louvain community detection at each resolution parameter) and calculated modularity-weighted z-rand scores across a range of resolution parameters, 0.75 to 1.1 at increments of 0.05. We selected local modularity-weighted z-rand peaks as solutions for further exploration.

## Connection diversity in the default mode network

The participation coefficient, which quantifies the degree to which a node is connected to a diverse set of communities, was calculated using the tools from the brain connectivity toolbox (https://sites.google.com/site/bctnet/). Since the participation coefficient calculation uses node strength (the sum of the connectivity weights across the network) in the denominator of its calculation, nodes with unusually low strength can produce unstably large participation coefficient values. We, thus, excluded regions with node strength in the bottom 25% of the network hubs. Participation coefficient calculations were performed across a range of

network densities from 5% to 20% at 1% intervals. These methods for calculating participation coefficient are in line with best practices in the literature [60,62,65,136,137]. Whole network participation coefficient was calculated for each DMN region, using the whole-brain connectivity matrix and community labels generated from the whole-brain Louvain community detection. Within DMN, participation coefficient was also calculated for each DMN node, using only the connections within the DMN and using the DMN subnetwork labels. Participation coefficient values were compared between DMN subnetworks using a within-subjects mixed model using mixedlm from statsmodels in Python 3 [138].

## Hippocampal-cortical network connectivity

To identify hippocampal-cortical networks, we calculated FC between each hippocampal ROI and every cortical ROI in the brain. To identify average connectivity of the hippocampus to a given network within-subject, FC weights were averaged together across cortical nodes, based on community affiliation. A 1-sample $t$ test was performed to determine what networks were, on average, significantly connected to the hippocampus. Significance was set at $p < 0.05$, Bonferroni corrected. We examined the connectivity of the resulting significant networks and subnetworks to the hippocampus as a function of the long axis (anterior versus posterior) and hemisphere (left versus right) in a within-subjects model as described above.

ROI level FC difference between the anterior and posterior hippocampus was performed in CONN toolbox version 18b. Using the cortical ROIs that were a part of large-scale networks connected to the hippocampus, a within-subject analysis was performed, which contrasted FC in the left and right anterior hippocampus against the left and right posterior hippocampus. Significance was set at $p < 0.05$, family-wise error (FWE) corrected.

## Representational similarity analysis

Using an independent dataset collected during a memory-based decision-making task [71], we examined the representational profile of each ROI in the DMN subnetworks (see S1 Text for a detailed description of the task). If the hippocampal-cortical networks identified using resting-state FC are functionally meaningful, then regions within those networks should have similar representational profiles.

In brief, 22 participants in this dataset viewed a set of 8 grocery items one at a time, in a pre-scanning session, and learned through trial and error whether each grocery item was desirable to a hypothetical customer within a store context or not. In each trial, a grocery item appeared in the context of a grocery store. There were 4 different grocery stores, and for half of the items, the particular grocery store modulated their desirability and for the other half, the grocery store context was irrelevant to their desirability. For example, the apple at grocery store "A" may be desirable, but the apple at grocery store "B" may be undesirable, which makes apple's desirability context dependent. Alternatively, desirability of a carton of milk may be the same regardless or grocery store, which makes it context invariant. Overall preference averaged to 50% desirable for both context dependent and invariant. Further, of the 4 contexts, 50% shared the same preference rules. Following successful learning, participants were put in the scanner and shown the context and a food item, one at a time, and asked to remember whether the food item was desirable to the hypothetical customer at the store context based on their learning phase. Feedback was not provided during scanning. In the original study, Mizrak and colleagues [71] examined representational similarity between memory-based decisions depending on (a) shared features between grocery items such as being modulated by the context or not; and (b) shared features between store contexts such as having similar desirability for the same grocery items or not.

Here, using the RSA toolbox (https://www.mrc-cbu.cam.ac.uk/methods-and-resources/toolboxes/license/ [139]), pattern similarity was calculated between memory-based decision-making trials for each ROI using Pearson correlation, excluding (a) trials that occurred during the same scanning run; and (b) incorrect trials. Thus, fluctuations in pattern similarity would be driven by variations in shared features. Correlations were then calculated between ROIs to calculate the similarity of their representational profiles. We then examined the representational similarity within cortico-hippocampal subnetworks versus between using a paired *t* test to examine whether these networks identified using resting-state FC did indeed create functionally relevant communities.

To account for univariate effects on the multivariate analysis, we further calculated the mean BOLD activity within an ROI for each trial and correlated the trial-wise mean BOLD activity for each pair of ROIs to create an ROI-by-ROI activation similarity matrix. Using the activation similarity matrix, we regressed out the influence of univariate activity from the RDM similarity matrix and reran the statistical analysis contrasting within- versus between-network RDM similarity.

To account for the effects of spatial proximity between ROIs, we calculated the Euclidean distance between the centroids of each pair of ROIs to create an ROI-by-ROI distance matrix. Using the distance matrix, we regressed out the influence of spatial proximity from the RDM similarity matrix and reran the statistical analysis contrasting within- versus between-network RDM similarity.

## Cognitive characterization of hippocampal-cortical networks

We decoded cortico-hippocampal networks to determine what terms are most frequently descriptive of the spatial distribution of the networks using the repository data version 0.7 (https://github.com/neurosynth/neurosynth-data)) of Neurosynth (https://github.com/neurosynth/neurosynth) [73]. The meta-analytic activation of every term in the Neurosynth database was correlated with the binarized volumetric masks of the hippocampal-cortical networks. For each hippocampal-cortical network, we examined the top terms whose meta-analytic activation correlated most strongly with the spatial layout of the hippocampal-cortical network. We removed terms that related to anatomy (e.g., vmPFC), network (e.g., default mode), technique (e.g., independent component), and report terms associated with cognitive functioning (e.g., navigation). We also removed terms that were redundant with other terms (e.g., "autobiographic" and "autobiographical memory"). When removing redundant terms, the term that was more specific was retained (e.g., "autobiographical memory" retained in favor of "memory"); otherwise, the term with the higher correlation to the network was retained. Word cloud visualizations were made using wordcloud 1.7.0 (https://pypi.org/project/wordcloud/).

## Supporting information

**S1 Fig. Inflated cortical surface, colored according to community membership for community detection analyses performed on split-halves of the sample.** Attn, attention; DMN, default mode network; Front Par, frontoparietal; MTN, medial temporal network; Som-Mot, somatomotor.
(PDF)

**S2 Fig. (A)** Inflated cortical surface, colored according to community membership for the secondary WashU120 dataset (top) and primary UCD dataset (bottom). **(B)** Connectivity matrix reordered by community to demonstrate the community structure of the group-averaged brain from the Secondary WashU120 dataset (left) and primary UCD dataset (right). Colors along the axis demonstrate which rows/columns belong to a given community. Color bar

represents Fisher Z-transformed correlation values. Arrow highlights the temporal polar network in the WashU120 dataset that has overall low (near 0) whole-brain FC. Attn, attention; DMN, default mode network; FC, functional connectivity; Front Par, frontoparietal; MTN, medial temporal network; Som-Mot, somatomotor. TP, Temporal polar; UCD, University of California, Davis. Data can be found at https://github.com/ajbarn/hippo_nets.
(PDF)

**S3 Fig.** A. The tSNR on the ventral surface of an inflated brain for regions with low tSNR (tSNR < 30) for the primary (UCD) and secondary (WashU120) datasets. B. The t-statistics for the difference in tSNR between the primary and secondary datasets. Regions in blue are those that showed significantly lower tSNR in the secondary dataset compared to the primary dataset. tSNR, temporal signal-to-noise ratio; UCD, University of California, Davis. Data can be found at https://github.com/ajbarn/hippo_nets.
(PDF)

**S4 Fig. Boxplots displaying the path length between the DMN and each of the other networks (targets) following removal of a given nontarget network.** Data can be found at https://github.com/ajbarn/hippo_nets. DMN, default mode network.
(PDF)

**S5 Fig. Boxplots displaying the path length between the hippocampus and each of the other networks (targets) following removal of a given nontarget network.** Data can be found at https://github.com/ajbarn/hippo_nets.
(PDF)

**S6 Fig. Boxplot of average region-to-region representational profile similarity between each cortical-hippocampal network (labeled as Target) to itself and every other network in the brain (labeled along the x-axis) across the sample.** AUD, auditory; AT, anterior temporal; DAN1, dorsal attention network 1; DAN2, dorsal attention network 2; FP, frontoparietal; Lang, language; MP, medial prefrontal; MTN, medial temporal network; PM, posterior medial; SAL1, salience 1; SAL2, salience 2; SM, somatomotor; VIS, visual. Data can be found at https://github.com/ajbarn/hippo_nets.
(PDF)

**S1 Table. Mean of the average FC between the hippocampus and the regions within each cortical network across subjects with standard deviation in parenthesis.** FC represented as the Fisher z-transformed correlation between the functional time series between 2 regions. FC, functional connectivity.
(DOCX)

**S2 Table. Correlation values of spatial overlap between the cortico-hippocampal networks and meta-analytic maps for cognitive terms.**
(DOCX)

**S3 Table. Regions of interest in the HCP-MMP atlas 1.0 that are found in the MTN, AT subnetwork, PM subnetwork, and MP subnetwork, with their community label provided.** AT, anterior temporal; MP, medial prefrontal; MTN, medial temporal network; PM, posterior medial.
(DOCX)

**S1 Text. A description of the multivoxel pattern estimation for the task dataset used in the representational profile similarity analysis provided by Mizrak et al. [71].**
(DOCX)

## Acknowledgments

We thank Costin Tanase, Dennis Thompson, and the Imaging Research Center for their technical contributions. We thank the Dynamic Memory Lab (http://dml.ucdavis.edu) for consultation on experimental design and analysis and Sam Audrain for feedback on the manuscript.

**Disclaimers:** Any opinions, findings, and conclusions expressed in this material are those of the author(s) and do not necessarily reflect the views of the Office of Naval Research or the U.S. Department of Defense.

## Author Contributions

**Conceptualization:** Alexander J. Barnett, Walter Reilly, Halle R. Dimsdale-Zucker, Charan Ranganath.

**Data curation:** Alexander J. Barnett, Walter Reilly, Halle R. Dimsdale-Zucker, Eda Mizrak, Zachariah Reagh.

**Formal analysis:** Alexander J. Barnett, Walter Reilly, Eda Mizrak.

**Funding acquisition:** Charan Ranganath.

**Project administration:** Alexander J. Barnett, Halle R. Dimsdale-Zucker.

**Supervision:** Charan Ranganath.

**Visualization:** Alexander J. Barnett.

**Writing – original draft:** Alexander J. Barnett.

**Writing – review & editing:** Alexander J. Barnett, Walter Reilly, Halle R. Dimsdale-Zucker, Eda Mizrak, Zachariah Reagh, Charan Ranganath.

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
