## [Editor Report · Decision Letter 0]

2 Oct 2020

Dear Dr Barnett, 

Thank you for submitting your manuscript entitled "Organization of cortico-hippocampal networks in the human brain" for consideration as a Research Article by PLOS Biology.

Your manuscript has now been evaluated by the PLOS Biology editorial staff, as well as by an academic editor with relevant expertise, and I am writing to let you know that we would like to send your submission out for external peer review.

Please re-submit your manuscript within two working days, i.e. by Oct 06 2020 11:59PM.

Kind regards,

Gabriel Gasque, Ph.D.,

Senior Editor

PLOS Biology

---

## [Decision Letter · Decision Letter 1]

6 Nov 2020

Dear Dr Barnett,

Thank you very much for submitting your manuscript "Organization of cortico-hippocampal networks in the human brain" for consideration as a Research Article at PLOS Biology. Your manuscript has been evaluated by the PLOS Biology editors, by an Academic Editor with relevant expertise, and by three independent reviewers.

In light of the reviews (below), we will not be able to accept the current version of the manuscript, but we would welcome re-submission of a much-revised version that takes into account the reviewers' comments. We cannot make any decision about publication until we have seen the revised manuscript and your response to the reviewers' comments. Your revised manuscript is also likely to be sent for further evaluation by the reviewers.

We expect to receive your revised manuscript within 3 months. However, as detailed below, we expect a thorough revision. Therefore, we would be happy to discuss an extension if one is deemed necessary to address our editorial and reviewer comments.

**IMPORTANT - SUBMITTING YOUR REVISION**

Your revisions should address the specific points made by each reviewer. Having discussed these comments with the Academic Editor, we think three key issues need to be thoroughly and satisfactorily addressed: Reviewer 1's point that there needs to be further analyses exploring the division between DMN and MTL before the DMN subdivision analyses are tackled. Second, Reviewer 2 (and Reviewer 3 to a lesser degree) highlight a crucial need to unpack the RSA methods/results with further analyses to make clear novel claims about the cognitive relevance of these subnetworks for episodic memory. Editorially, we feel that this aspect of the study provides the most novel insight beyond past parcellation studies, so feel that such expansion would be needed for further consideration at PLOS Biology. Reviewer 2 makes a key point about controlling for potential spurious effects in your functional connectivity analyses due to spatial proximity. Finally, from an editorial perspective, we would also like to see the revision more clearly placing your novel findings within the existing literature on MTL and DMN parcellation studies, to allow our broad readership to understand the major new advances provided. We would also need to be reassured that your results and conclusions stand after performing the suggested methodological controls and additional analyses. 

Please submit the following files along with your revised manuscript:

*Re-submission Checklist*

*Published Peer Review*

*PLOS Data Policy*

*Blot and Gel Data Policy*

Sincerely,

Gabriel Gasque, Ph.D.,

Senior Editor,

ggasque@plos.org,

PLOS Biology

REVIEWS:

Reviewer #1: In this paper, Barnett et al. seek to clarify the relationship between medial temporal lobes (MTL) and default mode (DMN) subnetworks. Past work has identified multiple subnetworks of the MTL and hippocampus that underlie memory-guided behavior; somewhat similarly, data-driven analyses have parcellated the DMN. Seeking to unify these approaches, the current studies characterize cortico-hippocampal networks and their relationship to episodic memory. They find that portions of the DMN and medial temporal lobes segregate from the rest of the DMN to form a medial temporal network (MTN). The MTN then seems to be responsible for transferring information from the visual network to the DMN and hippocampus. The DMN was further subdivided into 3 smaller subnetworks, which partly vary in their connectivity with the hippocampus and their functionality.

Overall, I found the paper to be solid and interesting to read. These networks are important to understand, and the relationship to function presented at the end was clever. However, one of the first findings described in the paper is that a network partitioning (using a community detection algorithm) pulls out the DMN, but only in part. A subset of traditionally DMN regions (parahippocampal cortex, retrosplenial cortex, and perirhinal cortex) instead grouped with some MTL and parietal regions. Further analyses in the paper really hinge on this relatively novel subdivision, so I think it deserves more scrutiny. Would the same algorithms applied to different (perhaps publicly available) datasets produce the same divisions? On the 40 subjects used here, would other methods of extracting the DMN produce more traditional results? Similarly, what would happen if the analyses associated with Figure 3 were repeated, but combining the DMN and MTN? Or using a different definition of the DMN? Essentially, I think that the notion of the division between the DMN and the MTN needs to be more fully explored, before the subdivisions of the DMN are tackled. I think the major impact of the paper lies here, and it will bring it to a level above the (many) DMN parcellation papers that already exist.

Minor comments

There are so many abbreviations, I think it would help the reader enormously to cut back. DMN and MTL are fine, but some combination of PHC, PRC, MTN, PM, AT, MP should just be spelled out.

I found the analysis and accompanying description for Figure 2 very satisfying and interesting.

Line 193—lead should be led

Line 442—on should be of

Reviewer #2: Barnett and colleagues conducted a thorough examination of the resting state connectivity structure of human cortico-hippocampal networks. Using a sample of 40 participants with 25-minutes of resting state functional connectivity, alongside an array graph theory-based analyses, the authors report a series of interesting results regarding the organization of large-scale brain networks and their relationship to the hippocampus:

(1) A replication of previous studies highlighting a distinct medial temporal network (MTN), separate from the default mode network (DMN)

(2) The MTN's role in mediating communication between the DMN and visual network (VN) 

(3) Demonstrating the DMN is composed of separable subnetworks, with new insights regarding how these different subnetworks may mediate communication within and outside of the DMN

(4) The relationship between the hippocampal long-axis and these separable DMN subsystems

The authors also included additional task-data and existing meta-analytic tools (i.e., Neurosynth) to extend these findings to how these regions may represent psychologically-meaningful information content.

Overall, I found this manuscript very relevant to current theories of hippocampal-whole brain macro-level organization. The analyses were thorough. The writing was clear. It was a pleasure to read. 

I have a few points that I hope will further benefit the quality of this manuscript and its conclusions. 

Major comments: 

(1) Can spatial proximity explain the role of the MTN in mediating crosstalk between DMN and VN?

The analysis examining the role of the MTN in mediating communication between the DMN/hippocampus and VN is particularly interesting. I appreciate the control analyses the authors have conducted, but I wonder whether this effect might be explained by spatial proximity of the MTN to both the DMN/hippocampus and VN. In other words, is the MTN the most spatially proximal network to both the DMN/hippocampus and VN simultaneously? Could spatial proximity/euclidean distance alone explain the observed path length effect? Or, is it something more about the underlying anatomy/functional coupling of these specific regions?

Given that FC tends to have sharp boundaries, it's unlikely that spatial proximity alone could account for this effect. But including a control analysis that examines this possibility can further clarify why the MTN should mediate communication between the DMN/HPC and VN. 

First, the authors can calculate the spatial proximity of each ROI to each other, and then average within network. [Note: the authors already did this as a control for the RSA analysis]. Is the MTN the network that is simultaneously closest to both DMN and VN/HPC and VN? If so, a potential control analysis could be to redo the 1000 resample control where 30 ROIs were randomly removed, but add an additional constraint that the 30 need to have at least the same (or some relatively comparable level) spatial proximity to both the DMN and VN as the MTN. This way, in addition to number of nodes in a network, the authors can control for the euclidean distance between the ROIs and the two target networks they purportedly mediate. 

Also, I would be curious to hear a bit more discussion re: why the MTN should mediate between the DMN and VN in the first place. For example, recent work has highlighted a relationship between the hippocampus/surrounding medial temporal lobes and the visual system (Shen et al., 2016; Ryan et al., 2020) - (how) does this finding extend this line of inquiry? Alternatively, the authors mention the idea that the MTN might represent "content" while regions of the DMN instead may represent "process" (or abstract, schema-like information). How would this fit with previous studies reporting pattern reinstatement of representations from encoding during recall, often specifically within DMN (in both AT and PM subnetworks, I believe) (e.g., Thakral, Wang & Rugg, 2017; Chen et al., 2017)?

(2) How generalizable/reliable is the reported network parcellation/community-breakdown? 

The authors parcellation of whole-brain networks generally replicates existing work. However, there were a few important differences. For example, both PRC and PHC were part of a common MTN network, rather than the originally proposed AT and PM systems (e.g., Ranganath & Ritchey, 2012). Also, earlier breakdowns of DMN connectivity described a core-DMN subsystem, consisting of the mPFC and PCC, which was not observed here (e.g., Andrews-Hanna et al., 2010). 

As a reader, I am not sure how much weight I should be putting into interpreting these findings - despite their theoretical importance (e.g., for PMAT). For one, the present study was based on 45 subjects (though, 25-minute scanning), and the group-averaged functional connectivity matrix was submitted to the community detection algorithm. Considering this study was based on resting state, is there a reason not to replicate this parcellation in a separate sample of subjects (e.g., using a publicly available dataset) to confirm the generalizability of the reported network parcellation? Or, perhaps, calculating the split-half reliability of these network breakdown within the existing sample could be another possibility? Also, from what I understand, reliable resting state network estimation can be achieved with considerably shorter scans (e.g., Birn et al., 2013), so perhaps splitting subjects' scans in half and replicating the network parcellation could be another possibility. In any case, I would like to hear the authors thoughts on the issue of interpreting the specific reported network breakdown, without a good sense of out-of-sample generalizability or within-sample reliability. 

(3) Why do regions within the same network have similar representational profiles? 

I really appreciate the inclusion of this dataset. I think it adds quite a bit to the resting state results. However, the lack of detail regarding experimental procedures and what made up the RSA matrix made it hard to fully understand. The Methods section mentions that more details can be found in the Supplementary Information, but I didn't seem to find any there…(unless I somehow missed it?). 

It would be helpful to get more details about the RDM used from the RSA analyses. I understand that there were 8 grocery items and 4 different grocery stores. So, was the RDM a 32 x 32 matrix, based on each combination of item and store? Or was it just a trial x trial similarity matrix? In any case, a schematic of the RDM in the Supplementary could be really helpful for readers in their attempts to parse this analysis and its conclusions. 

Also, I'm quite curious as to why the authors observed more similar 'representation profiles' in ROIs within as opposed to across networks. Can we leverage the experimental paradigm to get a bit more insight into what psychological features drive this result? 

For example, to go back to the MTN (point #1), the authors argue that the MTN may "provide [a] high-level feature representation of the external world" (p 18, ln 394). Can we find any evidence for that in the RSA analysis? For example, are representations more fine-grained in the VN than MTN than DMN? Couldn't this be quantified to some extent by looking at the off-diagonal variability in the ROI-level RSA matrices (more distinct patterns across trials, more fine-grained neural representations)? It's not perfect, as regions with noisier patterns of multivoxel activity might also produce this effect, but it seems in line with the authors' predictions. 

Alternatively, if the RDM was based on objects (grocery items) and contexts (grocery stores), could the authors test whether context or item drives the similarity effects in these different networks? The authors' thoughts on what is actually driving the representational similarity would be appreciated and could benefit some predictions made in the manuscript. 

Minor comments: 

Page 6, Line 139: "mixed connectivity"

- What does "mixed connectivity" mean? Across-subjects? If so, perhaps a ratio of subjects that showed these effects would be helpful.

Page 7, Line 147: "higher-order feature encoding" 

- I'm not sure what this means. Some additional elaboration could be helpful.

Page 10-11: Participation coefficients

- General question - what's the difference between these participation coefficient analyses and the network-holdout style analyses? In other words, does a high within-DMN participation coefficient imply that removal will lead to an overall increase in path length as compared to a network with lower within-DMN participation? In general, the participation coefficient analyses seemed to not have too much background (relative to the other analyses). Adding a sentence at the beginning of this section on why this matters could be helpful. 

- "whereas nodes in the AT subnetwork and PM subnetwork may interface with networks outside of the DMN." Did the PM have a significantly higher participation coefficient than the MP when looking at extra-DMN connections? Was that comparison reported?

Figure 7

- Including the network labels here, on the figure itself, would be helpful. 

Figure 8

- The ROIs (on the left side) should have labels on the figure. It would make it more immediately legible. 

References: 

Andrews-Hanna, J. R., Reidler, J. S., Sepulcre, J., Poulin, R., & Buckner, R. L. (2010). Functional-anatomic fractionation of the brain's default network. Neuron, 65(4), 550-562.

Birn, R. M., Molloy, E. K., Patriat, R., Parker, T., Meier, T. B., Kirk, G. R., ... & Prabhakaran, V. (2013). The effect of scan length on the reliability of resting-state fMRI connectivity estimates. Neuroimage, 83, 550-558.

Chen, J., Leong, Y. C., Honey, C. J., Yong, C. H., Norman, K. A., & Hasson, U. (2017). Shared memories reveal shared structure in neural activity across individuals. Nature neuroscience, 20(1), 115-125.

Ryan, J. D., Shen, K., & Liu, Z. X. (2020). The intersection between the oculomotor and hippocampal memory systems: empirical developments and clinical implications. Annals of the New York Academy of Sciences, 1464(1), 115.

Shen, K., Bezgin, G., Selvam, R., McIntosh, A. R., & Ryan, J. D. (2016). An anatomical interface between memory and oculomotor systems. Journal of Cognitive Neuroscience, 28(11), 1772-1783.

Thakral, P. P., Wang, T. H., & Rugg, M. D. (2017). Decoding the content of recollection within the core recollection network and beyond. Cortex, 91, 101-113.

Reviewer #3: In this paper, Barnett and colleagues provide an updated framework for understanding how resting state networks are organized, and how they interact with the hippocampus. This work takes a data-driven approach to investigate the well-studied "default mode network" or DMN, and illustrates that it can be 1) differentiated from a "medial temporal lobe network" (MTN) and 2) can itself be split up into three sub-networks. I reviewed this paper before for another journal and I was pleased to see how the authors have incorporated the previous reviewers' suggestions and updated the manuscript accordingly. I would like to commend the authors for making the connection between the current work and the previous PMAT framework much more clear. I have only a few minor comments.

1) The part of the paper that confused me the most, as a relative outsider to the graph theoretical world was the section on lines 217-245. These results made slightly more sense to me after reading the discussion (lines 465-475), but perhaps some of these analyses could be better motivated up front so that the reader can better appreciate the importance of these results (and the interpretation of Figure 4). 

2) Figure 6 depicts the representational similarity analysis results for the DMN sub-networks and the MTN. I am curious how these similarity values change when you compare the similarity between the MTN and DMN seeds and the other resting state networks (e.g. Visual, language, etc). I am assuming that similarity would be lower, but are they significantly lower than the ones included/plotted? For completeness this could be included in a supplemental figure or table. 

3) In the legend of Figure 2, there are color keys for the anterior and posterior hippocampus, but I don't see these colors in this particular figure. Am I missing something or should those colors be removed?

4) On line 133, it is stated that: "The posterior hippocampus was defined as all the hippocampus posterior to the last slice of the uncal notch." (also in Methods on lines 632 and 633). I believe that the authors meant the "uncal apex" as described by Poppenk et al., (2013).

---

## [Decision Letter · Decision Letter 2]

23 Apr 2021

Dear Dr Barnett,

Thank you for submitting your revised Research Article entitled "Organization of cortico-hippocampal networks in the human brain" for publication in PLOS Biology. I have now obtained advice from the original reviewers and have discussed their comments with the Academic Editor. 

Based on the reviews, we will probably accept this manuscript for publication; however, we would like you to dedicate a few sentences in the Discussion to relating your DMN/MTN findings to the hippocampal contextual processing findings in the Mizrak et al., 2019 bioRxiv preprint (Ref 71). In addition, you would need to satisfactorily address the data and other policy-related requests listed below my signature.

We expect to receive your revised manuscript within two weeks. 

*Published Peer Review History*

*Early Version*

Sincerely,

Gabriel Gasque, Ph.D.,

Senior Editor,

ggasque@plos.org,

PLOS Biology

TITLE:

We would like to suggest the following title, which we think might be of more interest to a broad readership:

"Organization of cortico-hippocampal networks in the human brain during memory-guided decision making."

Do let us know if you think this suggestion misrepresents your findings, and we would be happy to work together with you on an alternative. 

ETHICS STATEMENT:

-- Please indicate within your manuscript whether your experiments were conducted according to the principles expressed in the Declaration of Helsinki or any other specific national or international ethical guidelines.

DATA POLICY:

Thank you for uploading your data to Github.

--Could you please update your README file to indicate how you analyzed the data to generate the quantitative plots displayed in the main and supporting Figures?

--Please also ensure that each figure legend in your manuscript includes information on where the underlying data can be found (https://github.com/ajbarn/hippo_nets).

BLURB:

Please provide one in our submission system.

DATA NOT SHOWN?

Reviewer remarks:

Reviewer #1: The authors have addressed my concerns. Nice job!

Reviewer #2: The authors have answered all of my questions and concerns. Their split-half reliability analysis, the external validation in a new sample and their control analysis for distance are important additions to the final manuscript (which I believe will make an important contribution to the field). I was not able to see the updated figures for this revision, as they did not seem to be attached. 

Reviewer #3: I am happy with the authors' responses and I am looking forward to seeing the acceptance of the revised manuscript.

---

## [Editor Report · Decision Letter 3]

7 May 2021

Dear Dr Barnett,

On behalf of my colleagues and the Academic Editor, Raphael Kaplan, I am pleased to say that we can in principle offer to publish your Research Article "Intrinsic connectivity reveals functionally distinct cortico-hippocampal networks in the human brain" in PLOS Biology, provided you address any remaining formatting and reporting issues. These will be detailed in an email that will follow this letter and that you will usually receive within 2-3 business days, during which time no action is required from you. Please note that we will not be able to formally accept your manuscript and schedule it for publication until you have made the required changes.

PRESS

Thank you again for supporting Open Access publishing. We look forward to publishing your paper in PLOS Biology. 

Sincerely, 

Gabriel Gasque, Ph.D. 

Senior Editor 

PLOS Biology